# TOWARD TRAINABILITY OF QUANTUM NEURAL NETWORKS

## ABSTRACT

Quantum Neural Networks (QNNs) have been recently proposed as generalizations of classical neural networks to achieve the quantum speed-up. Despite the potential to outperform classical models, serious bottlenecks exist for training QNNs; namely, QNNs with random structures have poor trainability due to the vanishing gradient with rate exponential to the input qubit number. The vanishing gradient could seriously influence the applications of large-size QNNs. In this work, we provide a first viable solution with theoretical guarantees. Specifically, we prove that QNNs with tree tensor and step controlled architectures have gradients that vanish at most polynomially with the qubit number. Moreover, our result holds irrespective of which encoding methods are employed. We numerically demonstrate QNNs with tree tensor and step controlled structures for the application of binary classification. Simulations show faster convergent rates and better accuracy compared to QNNs with random structures.

## 1 INTRODUCTION

Neural Networks (Hecht-Nielsen, 1992) using gradient-based optimizations have dramatically advanced researches in discriminative models, generative models, and reinforcement learning. To efficiently utilize the parameters and practically improve the trainability, neural networks with specific architectures (LeCun et al., 2015) are introduced for different tasks, including convolutional neural networks (Krizhevsky et al., 2012) for image tasks, recurrent neural networks (Zaremba et al., 2014) for the time series analysis, and graph neural networks (Scarselli et al., 2008) for tasks related to graph-structured data. Recently, the neural architecture search (Elsken et al., 2019) is proposed to improve the performance of the networks by optimizing the neural structures.

Despite the success in many fields, the development of the neural network algorithms could be limited by the large computation resources required for the model training. In recent years, quantum computing has emerged as one solution to this problem, and has evolved into a new interdisciplinary field known as the quantum machine learning (QML) (Biamonte et al., 2017; Havlíček et al., 2019). Specifically, variational quantum circuits (Benedetti et al., 2019) have been explored as efficient protocols for quantum chemistry (Kandala et al., 2017) and combinatorial optimizations (Zhou et al., 2018). Compared to the classical circuit models, quantum circuits have shown greater expressive power (Du et al., 2020a), and demonstrated quantum advantage for the low-depth case (Bravyi et al., 2018). Due to the robustness against noises, variational quantum circuits have attracted significant interest for the hope to achieve the quantum supremacy on near-term quantum computers (Arute et al., 2019).

Quantum Neural Networks (QNNs) (Farhi & Neven, 2018; Schuld et al., 2020; Beer et al., 2020) are the special kind of quantum-classical hybrid algorithms that run on trainable quantum circuits. Recently, small-scale QNNs have been implemented on real quantum computers (Havlíček et al., 2019) for supervised learning tasks. The training of QNNs aims to minimize the objective function $f$ with respect to parameters $\boldsymbol{\theta}$. Inspired by the classical optimizations of neural networks, a natural strategy to train QNNs is to exploit the gradient of the loss function (Crooks, 2019). However, the recent work (McClean et al., 2018) shows that $n$-qubit quantum circuits with random structures and large depth $L = \mathcal{O}(\text{poly}(n))$ tend to be approximately unitary 2-design (Harrow & Low, 2009), and the partial derivative vanishes to zero exponentially with respect to $n$. The vanishing gradient problem is usually referred to as the Barren Plateaus (McClean et al., 2018), and could affect the

trainability of QNNs in two folds. Firstly, simply using the gradient-based method like Stochastic Gradient Descent (SGD) to train the QNN takes a large number of iterations. Secondly, the estimation of the derivatives needs an extremely large number of samples from the quantum output to guarantee a relatively accurate update direction (Chen et al., 2018). To avoid the Barren Plateaus phenomenon, we explore QNNs with special structures to gain fruitful results.

In this work, we introduce QNNs with special architectures, including the tree tensor (TT) structure (Huggins et al., 2019) referred to as TT-QNNs and the setp controlled structure referred to as SC-QNNs. We prove that for TT-QNNs and SC-QNNs, the expectation of the gradient norm of the objective function is bounded.

**Theorem 1.1.** *(Informal) Consider the $n$-qubit TT-QNN and the $n$-qubit SC-QNN defined in Figure 1-2 and corresponding objective functions $f_{TT}$ and $f_{SC}$ defined in (3-4), then we have:*

$$\frac{1 + \log n}{2n} \cdot \alpha(\rho_{in}) \leq \mathbb{E}_{\boldsymbol{\theta}} \|\nabla_{\boldsymbol{\theta}} f_{TT}\|^2 \leq 2n - 1,$$

$$\frac{1 + n_c}{2^{1+n_c}} \cdot \alpha(\rho_{in}) \leq \mathbb{E}_{\boldsymbol{\theta}} \|\nabla_{\boldsymbol{\theta}} f_{SC}\|^2 \leq 2n - 1,$$

*where $n_c$ is the number of CNOT operations that directly link to the first qubit channel in the SC-QNN, the expectation is taken for all parameters in $\boldsymbol{\theta}$ with uniform distributions in $[0, 2\pi]$, and $\alpha(\rho_{in}) \geq 0$ is a constant that only depends on the input state $\rho_{in} \in \mathbb{C}^{2^n \times 2^n}$. Moreover, by preparing $\rho_{in}$ using the $L$-layer encoding circuit in Figure 4, the expectation of $\alpha(\rho_{in})$ could be further lower bounded as $\mathbb{E}\alpha(\rho_{in}) \geq 2^{-2L}$.*

Compared to random QNNs with $2^{-\mathcal{O}(\text{poly}(n))}$ derivatives, the gradient norm of TT-QNNs ad SC-QNNs is greater than $\Omega(1/n)$ or $\Omega(2^{-n_c})$ that could lead to better trainability. Our contributions are summarized as follows:

- We prove $\tilde{\Omega}(1/n)$ and $\tilde{\Omega}(2^{-n_c})$ lower bounds on the expectation of the gradient norm of TT-QNNs and SC-QNNs, respectively, that guarantees the trainability on related optimization problems. Our theorem does not require the unitary 2-design assumption in existing works and is more realistic to near-term quantum computers.

- We prove that by employing the encoding circuit in Figure 4 to prepare $\rho_{\text{in}}$, the expectation of term $\alpha(\rho_{\text{in}})$ is lower bounded by a constant $2^{-2L}$. Thus, we further lower bounded the expectation of the gradient norm to the term independent from the input state.

- We simulate the performance of TT-QNNs, SC-QNNs, and random structure QNNs on the binary classification task. All results verify proposed theorems. Both TT-QNNs and SC-QNNs show better trainability and accuracy than random QNNs.

Our proof strategy could be adopted for analyzing QNNs with other architectures as future works. With the proven assurance on the trainability of TT-QNNs and SC-QNNs, we eliminate one bottleneck in front of the application of large-size Quantum Neural Networks.

The rest parts of this paper are organized as follows. We address the preliminary including the definitions, the basic quantum computing knowledge and related works in Section 2. The QNNs with special structures and the corresponding results are presented in Section 3. We implement the binary classification using QNNs with the results shown in Section 4. We make conclusions in Section 5.

## 2 PRELIMINARY

### 2.1 NOTATIONS AND THE BASIC QUANTUM COMPUTING

We use $[N]$ to denote the set $\{1, 2, \cdots, N\}$. The form $\|\cdot\|$ denotes the $\|\cdot\|_2$ norm for vectors. We denote $a_j$ as the $j$-th component of the vector $\boldsymbol{a}$. The tensor product operation is denoted as "$\otimes$". The conjugate transpose of a matrix $A$ is denoted as $A^\dagger$. The trace of a matrix $A$ is denoted as $\text{Tr}[A]$. We denote $\nabla_{\boldsymbol{\theta}} f$ as the gradient of the function $f$ with respect to the vector $\boldsymbol{\theta}$. We employ notations $\mathcal{O}$ and $\tilde{\mathcal{O}}$ to describe the standard complexity and the complexity ignoring minor terms, respectively.

Now we introduce the quantum computing. The pure state of a qubit could be written as $|\phi\rangle = a|0\rangle + b|1\rangle$, where $a, b \in \mathbb{C}$ satisfies $|a|^2 + |b|^2 = 1$, and $\{|0\rangle = (1,0)^T, |1\rangle = (0,1)^T\}$, respectively. The $n$-qubit space is formed by the tensor product of $n$ single-qubit spaces. For the vector $\boldsymbol{x} \in \mathbb{R}^{2^n}$, the amplitude encoded state $|\boldsymbol{x}\rangle$ is defined as $\frac{1}{\|\boldsymbol{x}\|} \sum_{j=1}^{2^n} x_j |j\rangle$. The dense matrix is defined as $\rho = |\boldsymbol{x}\rangle\langle\boldsymbol{x}|$ for the pure state, in which $\langle\boldsymbol{x}| = (|\boldsymbol{x}\rangle)^\dagger$. A single-qubit operation to the state behaves like the matrix-vector multiplication and can be referred to as the gate —□— in the quantum circuit language. Specifically, single-qubit operations are often used including $R_X(\theta) = e^{-i\theta X}$, $R_Y(\theta) = e^{-i\theta Y}$, and $R_Z(\theta) = e^{-i\theta Z}$:

$$X = \begin{pmatrix} 0 & 1 \\ 1 & 0 \end{pmatrix}, Y = \begin{pmatrix} 0 & -i \\ i & 0 \end{pmatrix}, Z = \begin{pmatrix} 1 & 0 \\ 0 & -1 \end{pmatrix}.$$

Pauli matrices $\{I, X, Y, Z\}$ will be referred to as $\{\sigma_0, \sigma_1, \sigma_2, \sigma_3\}$ for the convenience. Moreover, two-qubit operations, the CNOT gate and the CZ gate, are employed for generating quantum entanglement:

$$\text{CNOT} = \text{—●—} = |0\rangle\langle0| \otimes \sigma_0 + |1\rangle\langle1| \otimes \sigma_1, \ \text{CZ} = \text{—●—} = |0\rangle\langle0| \otimes \sigma_0 + |1\rangle\langle1| \otimes \sigma_3.$$

We could obtain information from the quantum system by performing measurements, for example, measuring the state $|\phi\rangle = a|0\rangle + b|1\rangle$ generates 0 and 1 with probability $p(0) = |a|^2$ and $p(1) = |b|^2$, respectively. Such a measurement operation could be mathematically referred to as calculating the average of the observable $O = \sigma_3$ under the state $|\phi\rangle$:

$$\langle\sigma_3\rangle_{|\phi\rangle} \equiv \langle\phi|\sigma_3|\phi\rangle \equiv \text{Tr}[|\phi\rangle\langle\phi| \cdot \sigma_3] = |a|^2 - |b|^2 = p(0) - p(1) = 2p(0) - 1.$$

The average of a unitary observable under arbitrary states is bounded by $[-1, 1]$.

## 2.2 RELATED WORKS

The barren plateaus phenomenon in QNNs is first noticed by McClean et al. (2018). They prove that for $n$-qubit random quantum circuits with depth $L = \mathcal{O}(\text{poly}(n))$, the expectation of the derivative to the objective function is zero, and the variance of the derivative vanishes to zero with rate exponential in the number of qubits $n$. Later, Cerezo et al. (2020) prove that for $L$-depth quantum circuits consisting of 2-design gates, the gradient with local observables vanishes with the rate $\mathcal{O}(2^{-O(L)})$. The result implies that in the low-depth $L = \mathcal{O}(\log n)$ case, the vanishing rate could be $\mathcal{O}(\frac{1}{\text{poly} n})$, which is better than previous exponential results. Recently, some techniques have been proposed to address the barren plateaus problem, including the special initialization strategy (Grant et al., 2019) and the layerwise training method (Skolik et al., 2020). We remark that these techniques rely on the assumption of low-depth quantum circuits. Specifically, Grant et al. (2019) initialize parameters such that the initial quantum circuit is equivalent to an identity matrix ($L = 0$). Skolik et al. (2020) train parameters in subsets in each layer, so that a low-depth circuit is optimized during the training of each subset of parameters.

Since random quantum circuits tend to be approximately unitary 2-design[1] as the circuit depth increases (Harrow & Low, 2009), and 2-design circuits lead to exponentially vanishing gradients (McClean et al., 2018), the natural idea is to consider circuits with special structures. On the other hand, tensor networks with hierarchical structures have been shown an inherent relationship with classical neural networks (Liu et al., 2019; Hayashi et al., 2019). Recently, quantum classifiers using hierarchical structure QNNs have been explored (Grant et al., 2018), including the tree tensor network and the multi-scale entanglement renormalization ansatz. Besides, QNNs with dissipative layers have shown the ability to avoid the barren plateaus (Beer et al., 2020). However, theoretical analysis of the trainability of QNNs with certain layer structures has been little explored (Sharma et al., 2020). Also, the 2-design assumption in the existing theoretical analysis (McClean et al., 2018; Cerezo et al., 2020; Sharma et al., 2020) is hard to implement exactly on near-term quantum devices.

## 3 QUANTUM NEURAL NETWORKS

In this section, we discuss the quantum neural networks in detail. Specifically, the optimizing of QNNs is presented in Section 3.1. We analyze QNNs with special structures in Section 3.2. We

---

[1]We refer the readers to Appendix B for a short discussion about the unitary 2-design.

introduce an approximate quantum input model in Section 3.3 which helps for deriving further theoretical bounds.

## 3.1 THE OPTIMIZING OF QUANTUM NEURAL NETWORKS

In this subsection, we introduce the gradient-based strategy for optimizing QNNs. Like the weight matrix in classical neural networks, the QNN involves a parameterized quantum circuit that mathematically equals to a parameterized unitary matrix $V(\boldsymbol{\theta})$. The training of QNNs aims to optimize the function $f$ defined as:

$$f(\boldsymbol{\theta}; \rho_{\text{in}}) = \frac{1}{2} + \frac{1}{2}\text{Tr}\left[O \cdot V(\boldsymbol{\theta}) \cdot \rho_{\text{in}} \cdot V(\boldsymbol{\theta})^{\dagger}\right] = \frac{1}{2} + \frac{1}{2}\langle O \rangle_{V(\boldsymbol{\theta}),\rho_{\text{in}}}, \tag{1}$$

where $O$ denotes the quantum observable and $\rho_{\text{in}}$ denotes the density matrix of the input quantum state. Generally, we could deploy the parameters $\boldsymbol{\theta}$ in a tunable quantum circuit arbitrarily. A practical tactic is to encode parameters $\{\theta_j\}$ as the phases of the single-qubit gates $\{e^{-i\theta_j\sigma_k}, k \in \{1, 2, 3\}\}$ while employing two-qubit gates $\{\text{CNOT, CZ}\}$ among them to generate quantum entanglement. This strategy has been frequently used in existing quantum circuit algorithms (Schuld et al., 2020; Benedetti et al., 2019; Du et al., 2020b) since the model suits the noisy near-term quantum computers. Under the single-qubit phase encoding case, the partial derivative of the function $f$ could be calculated using the parameter shifting rule (Crooks, 2019),

$$\frac{\partial f}{\partial \theta_j} = \frac{1}{2}\langle O \rangle_{V(\boldsymbol{\theta}_+),\rho_{\text{in}}} - \frac{1}{2}\langle O \rangle_{V(\boldsymbol{\theta}_-),\rho_{\text{in}}} = f(\boldsymbol{\theta}_+; \rho_{\text{in}}) - f(\boldsymbol{\theta}_-; \rho_{\text{in}}), \tag{2}$$

where $\boldsymbol{\theta}_+$ and $\boldsymbol{\theta}_-$ are different from $\boldsymbol{\theta}$ only at the $j$-th parameter: $\theta_j \to \theta_j \pm \frac{\pi}{4}$. Thus, the gradient of $f$ could be obtained by estimating quantum observables, which allows employing quantum computers for fast optimizations using stochastic gradient descents.

## 3.2 QUANTUM NEURAL NETWORKS WITH SPECIAL STRUCTURES

In this subsection, we introduce quantum neural networks with tree tensor (TT) (Grant et al., 2018) and step controlled (SC) architectures. We prove that the expectation of the square of gradient $\ell_2$-norm for the TT-QNN and the SC-QNN are lower bounded by $\tilde{\Omega}(1/n)$ and $\tilde{\Omega}(2^{-n_c})$, respectively, where $n_c$ is a parameter in the SC-QNN that is independent from the qubit number $n$. Moreover, the corresponding theoretical analysis does not rely on 2-design assumptions for quantum circuits.

Now we discuss proposed quantum neural networks in detail. We consider the $n$-qubit QNN constructed by the single-qubit gate $W_j^{(k)} = e^{-i\theta_j^{(k)}\sigma_2}$ and the CNOT gate $\sigma_1 \otimes |1\rangle\langle 1| + \sigma_0 \otimes |0\rangle\langle 0|$. We define the $k$-th parameter in the $j$-th layer as $\theta_j^{(k)}$. We only employ $R_Y$ rotations for single-qubit gates, due to the fact that the real world data lie in the real space, while applying $R_X$ and $R_Z$ rotations would introduce imaginary term to the quantum state.

We demonstrate the TT-QNN in Figure 1 for the $n = 4$ case, which employs CNOT gates in the binary tree form to achieve the quantum entanglement. The circuit of the SC-QNN could be divided into two parts: in the first part, CNOT operations are performed between adjacent qubit channels; in the second part, CNOT operations are performed between different qubit channels and the first qubit channel. An illustration of the SC-QNN is shown in Figure 2 for the $n = 4$ and $n_c = 2$ case, where $n_c$ denotes the number of CNOT operations that directly link to the first qubit channel. The number of parameters in both the TT-QNN and the SC-QNN are $2n - 1$. We consider correspond objective functions that are defined as

$$f_{\text{TT}}(\boldsymbol{\theta}) = \frac{1}{2} + \frac{1}{2}\text{Tr}[\sigma_3 \otimes I^{\otimes(n-1)}V_{\text{TT}}(\boldsymbol{\theta})\rho_{\text{in}}V_{\text{TT}}(\boldsymbol{\theta})^{\dagger}], \tag{3}$$

$$f_{\text{SC}}(\boldsymbol{\theta}) = \frac{1}{2} + \frac{1}{2}\text{Tr}[\sigma_3 \otimes I^{\otimes(n-1)}V_{\text{SC}}(\boldsymbol{\theta})\rho_{\text{in}}V_{\text{SC}}(\boldsymbol{\theta})^{\dagger}], \tag{4}$$

where $V_{\text{TT}}(\boldsymbol{\theta})$ and $V_{\text{SC}}(\boldsymbol{\theta})$ denotes the parameterized quantum circuit operations for the TT-QNN and the SC-QNN, respectively. We employ the observable $\sigma_3 \otimes I^{\otimes(n-1)}$ in Eq. (3-4) such that objective functions could be easily estimated by measuring the first qubit in corresponding quantum circuits.

Main results of this section are stated in Theorem 3.1, in which we prove $\tilde{\Omega}(1/n)$ and $\tilde{\Omega}(2^{-n_c})$ lower bounds on the expectation of the square of the gradient norm for the TT-QNN and the SC-QNN, respectively. By setting $n_c = \mathcal{O}(\log n)$, we could obtain the linear inverse bound for the SC-QNN as well. We provide the proof of Theorem 3.1 in Appendix D and Appendix G.

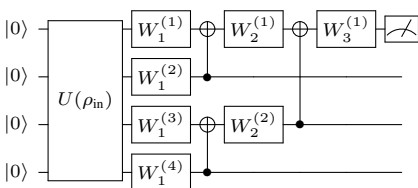

Figure 1: Quantum Neural Network with the Tree Tensor structure ($n = 4$).

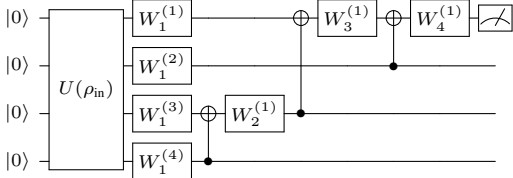

Figure 2: Quantum Neural Network with the Step Controlled structure ($n = 4$, $n_c = 2$).

**Theorem 3.1.** *Consider the $n$-qubit TT-QNN and the $n$-qubit SC-QNN defined in Figure 1-2 and corresponding objective functions $f_{TT}$ and $f_{SC}$ defined in Eq. (3-4), then we have:*

$$\frac{1 + \log n}{2n} \cdot \alpha(\rho_{in}) \leq \mathbb{E}_{\boldsymbol{\theta}} \|\nabla_{\boldsymbol{\theta}} f_{TT}\|^2 \leq 2n - 1, \tag{5}$$

$$\frac{1 + n_c}{2^{1+n_c}} \cdot \alpha(\rho_{in}) \leq \mathbb{E}_{\boldsymbol{\theta}} \|\nabla_{\boldsymbol{\theta}} f_{SC}\|^2 \leq 2n - 1, \tag{6}$$

*where $n_c$ is the number of CNOT operations that directly link to the first qubit channel in the SC-QNN, the expectation is taken for all parameters in $\boldsymbol{\theta}$ with uniform distributions in $[0, 2\pi]$, $\rho_{in} \in \mathbb{C}^{2^n \times 2^n}$ denotes the input state, $\alpha(\rho_{in}) = Tr\left[\sigma_{(1,0,\cdots,0)} \cdot \rho_{in}\right]^2 + Tr\left[\sigma_{(3,0,\cdots,0)} \cdot \rho_{in}\right]^2$, and $\sigma_{(i_1,i_2,\cdots,i_n)} \equiv \sigma_{i_1} \otimes \sigma_{i_2} \otimes \cdots \otimes \sigma_{i_n}$.*

From the geographic view, the value $\mathbb{E}_{\boldsymbol{\theta}} \|\nabla_{\boldsymbol{\theta}} f\|^2$ characterizes the global steepness of the function surface in the parameter space. Optimizing the objective function $f$ using gradient-based methods could be hard if the norm of the gradient vanishes to zero. Thus, lower bounds in Eq. (5-6) provide a theoretical guarantee on optimizing corresponding functions, which then ensures the trainability of QNNs on related machine learning tasks.

From the technical view, we provide a new theoretical framework during proving Eq. (5-6). Different from existing works (McClean et al., 2018; Grant et al., 2019; Cerezo et al., 2020) that define the expectation as the average of the finite unitary 2-design group, we consider the uniform distribution in which each parameter in $\boldsymbol{\theta}$ varies continuously in $[0, 2\pi]$. Our assumption suits the quantum circuits that encode the parameters in the phase of single-qubit rotations. Moreover, the result in Eq. (6) gives the first proven guarantee on the trainability of QNN with linear depth. Our framework could be extensively employed for analyzing QNNs with other different structures as future works.

### 3.3 PREPARE THE QUANTUM INPUT MODEL: A VARIATIONAL CIRCUIT APPROACH

State preparation is an essential part of most quantum algorithms, which encodes the classical information into quantum states. Specifically, the amplitude encoding $|\boldsymbol{x}\rangle = \sum_{i=1}^{2^n} x_i/\|\boldsymbol{x}\| |i\rangle$ allows storing the $2^n$-dimensional vector in $n$ qubits. Due to the dense encoding nature and the similarity to the original vector, the amplitude encoding is preferred as the state preparation by many QML algorithms (Harrow et al., 2009; Rebentrost et al., 2014; Kerenidis et al., 2019). Despite the wide application in quantum algorithms, efficient amplitude encoding remains little explored. Existing work (Park et al., 2019) could prepare the amplitude encoding state in time $\mathcal{O}(2^n)$ using a quantum circuit with $\mathcal{O}(2^n)$ depth, which is prohibitive for large-size data on near-term quantum computers. In fact, arbitrary quantum amplitude encoding with polynomial gate complexity remains an open problem.

---

**Algorithm 1** Quantum Input Model

---

**Require:** The input vector $\boldsymbol{x}_{\text{in}} \in \mathbb{R}^{2^n}$, the number of alternating layers $L$, the iteration time $T$, and the learning rate $\{\eta(t)\}_{t=0}^{T-1}$.
**Ensure:** A parameter vector $\boldsymbol{\beta}^*$ which tunes the approximate encoding circuit $U(\boldsymbol{\beta}^*)$.
 1: Initialize $\{\beta_j^{(k)}\}_{j,k=1}^{n,2L+1}$ randomly in $[0, 2\pi]$. Denote the parameter vector as $\boldsymbol{\beta}^{(0)}$.
 2: **for** $t \in \{0, 1, \cdots, T-1\}$ **do**
 3:     Run the circuit in Figure 3 classically to calculate the gradient $\nabla_{\boldsymbol{\beta}} f_{\text{input}}|_{\boldsymbol{\beta}=\boldsymbol{\beta}^{(t)}}$ using the parameter shifting rule (2), where the function $f_{\text{input}}$ is defined in (7).
 4:     Update the parameter $\boldsymbol{\beta}^{(t+1)} = \boldsymbol{\beta}^{(t)} - \eta(t) \cdot \nabla_{\boldsymbol{\beta}} f_{\text{input}}|_{\boldsymbol{\beta}=\boldsymbol{\beta}^{(t)}}$.
 5: **end for**
 6: Output the trained parameter $\boldsymbol{\beta}^*$.

---



Figure 3: The parameterized alternating layered circuit $W(\boldsymbol{\beta})$ ($n = 8$, $L = 1$) for training the corresponding encoding circuit of the input $\boldsymbol{x}_{\text{in}}$.

Figure 4: The encoding circuit $U(\boldsymbol{\beta}^*)$, where $W(\boldsymbol{\beta}^*)$ is the trained parameterized circuit in Figure 3.

In this subsection, we introduce a quantum input model for approximately encoding the arbitrary vector $\boldsymbol{x}_{\text{in}} \in \mathbb{R}^{2^n}$ in the amplitude of the quantum state $|\boldsymbol{x}_{\text{in}}\rangle$. The main idea is to classically train an alternating layered circuit as summarized in Algorithm 1 and Figures 3-4. Now we explain the detail of the input model. Firstly, we randomly initialize the parameter $\boldsymbol{\beta}^{(0)}$ in the circuit 3. Then, we train the parameter to minimize the objective function defined in (7) through the gradient descent,

$$f_{\text{input}}(\boldsymbol{\beta}) = \frac{1}{n} \sum_{i=1}^{n} \langle O_i \rangle_{W(\boldsymbol{\beta})|\boldsymbol{x}_{\text{in}}\rangle} = \frac{1}{n} \sum_{i=1}^{n} \frac{1}{\|\boldsymbol{x}_{\text{in}}\|^2} \text{Tr}[O_i \cdot W(\boldsymbol{\beta}) \cdot \boldsymbol{x}_{\text{in}} \boldsymbol{x}_{\text{in}}^T \cdot W(\boldsymbol{\beta})^\dagger], \qquad (7)$$

where $O_i = \sigma_0^{\otimes(i-1)} \otimes \sigma_3 \otimes \sigma_0^{\otimes(n-i)}, \forall i \in [n]$, and $W(\boldsymbol{\beta})$ denotes the tunable alternating layered circuit in Figure 3. Note that although the framework is given in the quantum circuit language, we actually calculate and update the gradient on classical computers by considering each quantum gate operation as the matrix multiplication. The output of Algorithm 1 is the trained parameter vector $\boldsymbol{\beta}^*$ which tunes the unitary $W(\boldsymbol{\beta}^*)$. The corresponding encoding circuit could then be implemented as $U(\boldsymbol{\beta}^*) = W(\boldsymbol{\beta}^*)^\dagger \cdot X^{\otimes n}$, which is low-depth and appropriate for near-term quantum computers. Structures of circuits $W$ and $U$ are illustrated in Figure 3 and 4, respectively. Suppose we could minimize the objective function (7) to $-1$. Then $\langle O_i \rangle = -1, \forall i \in [n]$, which means the final output state in Figure 3 equals to $W(\boldsymbol{\beta}^*)|\boldsymbol{x}_{\text{in}}\rangle = |1\rangle^{\otimes n}$.[2] Thus the state $|\boldsymbol{x}_{\text{in}}\rangle$ could be prepared exactly by applying the circuit $U(\boldsymbol{\beta}^*) = W(\boldsymbol{\beta}^*)^\dagger X^{\otimes n}$ on the state $|0\rangle^{\otimes n}$. However we could not always optimize the loss in Eq. (7) to $-1$, which means the framework could only prepare the amplitude encoding state approximately.

---

[2]We ignore the global phase on the quantum state. Based on the assumptions of this paper, all quantum states that encode input data lie in the real space, which then limit the global phase as 1 or -1. For the former case, nothing needs to be done; for the letter case, the global phase could be introduced by adding a single qubit gate $e^{-i\pi\sigma_0} = -\sigma_0 = -I$ on anyone among qubit channels.

---

**Algorithm 2** QNNs for the Binary Classification Training

---

**Require:** Quantum input states $\{\rho_i^{\text{train}}\}_{i=1}^S$ for the dataset $\{(\boldsymbol{x}_i^{\text{train}}, y_i)\}_{i=1}^S$, the quantum observable $O$, the parameterized quantum circuit $V(\boldsymbol{\theta})$, the iteration time $T$, the batch size $s$, and learning rates $\{\eta_\theta(t)\}_{t=0}^{T-1}$ and $\{\eta_b(t)\}_{t=0}^{T-1}$.
**Ensure:** The trained parameters $\boldsymbol{\theta}^*$ and $b^*$.
1: Initialize each parameter in $\boldsymbol{\theta}^{(0)}$ randomly in $[0, 2\pi]$ and initialize $b^{(0)} = 0$.
2: **for** $t \in \{0, 1, \cdots, T-1\}$ **do**
3:  Randomly sample an index subset $I_t \subset [S]$ with size $s$. Calculate the gradient $\nabla_{\boldsymbol{\theta}} \ell_{I_t}(\boldsymbol{\theta}, b)|_{\boldsymbol{\theta}, b=\boldsymbol{\theta}^{(t)}, b^{(t)}}$ and $\nabla_b \ell_{I_t}(\boldsymbol{\theta}, b)|_{\boldsymbol{\theta}, b=\boldsymbol{\theta}^{(t)}, b^{(t)}}$ using the chain rule and the parameter shifting rule (2), where the function $\ell_{I_t}(\boldsymbol{\theta}, b)$ is defined in (8).
4:  Update parameters $\boldsymbol{\theta}^{(t+1)}/b^{(t+1)} = \boldsymbol{\theta}^{(t)}/b^{(t)} - \eta_{\theta/b}(t) \cdot \nabla_{\boldsymbol{\theta}/b} \ell_{I_t}(\boldsymbol{\theta}, b)|_{\boldsymbol{\theta}, b=\boldsymbol{\theta}^{(t)}, b^{(t)}}$.
5: **end for**
6: Output trained parameters $\boldsymbol{\theta}^* = \boldsymbol{\theta}^{(T)}$ and $b^* = b^{(T)}$.

---

An interesting result is that by employing the encoding circuit in Figure 4 for constructing the input state in Section 3.2 as $\rho_{\text{in}} = U(\boldsymbol{\beta})(|0\rangle\langle 0|)^{\otimes n} U(\boldsymbol{\beta})^\dagger$, we could bound the expectation of $\alpha(\rho_{\text{in}})$ defined in Theorem 3.1 by a constant that only relies on the layer $L$ (Theorem 3.2).

**Theorem 3.2.** *Suppose the state $\rho_{in}$ is prepared by the L-layer encoding circuit in Figure 4, then we have,*

$$\mathbb{E}_{\boldsymbol{\beta}} \alpha(\rho_{in}) \geq 2^{-2L},$$

*where $\boldsymbol{\beta}$ denotes all variational parameters in the encoding circuit, and the expectation is taken for all parameters in $\boldsymbol{\beta}$ with uniform distributions in $[0, 2\pi]$.*

We provide the proof of Theorem 3.2 and more details about the input model in Appendix E. Theorem 3.2 can be employed with Theorem 3.1 to derive Theorem 1.1, in which the lower bound for the expectation of the gradient norm is independent from the input state.

## 4 APPLICATION: QNNs FOR THE BINARY CLASSIFICATION

### 4.1 QNNs: TRAINING AND PREDICTION

In this section, we show how to train QNNs for the binary classification in quantum computers. First of all, for the training and test data denoted as $\{(\boldsymbol{x}_i^{\text{train}}, y_i)\}_{i=1}^S$ and $\{(\boldsymbol{x}_j^{\text{test}}, y_j)\}_{j=1}^Q$, where $y_i \in \{0, 1\}$ denotes the label, we prepare corresponding quantum input states $\{\rho_i^{\text{train}}\}_{i=1}^S$ and $\{\rho_j^{\text{test}}\}_{j=1}^Q$ using the encoding circuit presented in Section 3.3. Then, we employ Algorithm 2 to train the parameter $\boldsymbol{\theta}$ and the bia $b$ via the stochastic gradient descent method for the given parameterized circuit $V$ and the quantum observable $O$. The parameter updating in each iteration is presented in the Step 3-4, which aims to minimize the loss defined in (8) for each input batch $I_t, \forall t \in [T]$:

$$\ell_{I_t}(\boldsymbol{\theta}) = \frac{1}{|I_t|} \sum_{i=1}^{|I_t|} \left( f(\boldsymbol{\theta}; \rho_i^{\text{train}}) - y_i + b \right)^2. \tag{8}$$

Based on the chain rule and the Eq. (2), derivatives $\frac{\partial \ell_{I_t}}{\partial \theta_j}$ and $\frac{\partial \ell_{I_t}}{\partial b}$ could be decomposed into products of objective functions $f$ with different variables, which could be estimated efficiently by counting quantum outputs. In practice, we calculate the value of the objective function by measuring the output state of the QNN for several times and averaging quantum outputs. After the training iteration we obtain the trained parameters $\boldsymbol{\theta}^*$ and $b^*$. Denote the quantum circuit $V^* = V(\boldsymbol{\theta}^*)$. We do test for an input state $\rho^{\text{test}}$ by calculating the objective function $f(\rho^{\text{test}}) = \frac{1}{2} + \frac{1}{2}\text{Tr}[O \cdot V^* \rho^{\text{test}} V^{*\dagger}]$. We classify the input $\rho^{\text{test}}$ as in the class 0 if $f(\rho^{\text{test}}) + b^* < \frac{1}{2}$, or the class 1 if $f(\rho^{\text{test}}) + b^* > \frac{1}{2}$.

The time complexity of the QNN training and test could be easily derived by counting resources for estimating all quantum observables. Denote the number of gates and parameters in the quantum circuit $V(\boldsymbol{\theta})$ as $n_{\text{gate}}$ and $n_{\text{para}}$, respectively. Denote the number of measurements for estimating each quantum observable as $n_{\text{train}}$ and $n_{\text{test}}$ for the training and test stages, respectively. Then, the time complexity to train QNNs is $\mathcal{O}(n_{\text{gate}} n_{\text{para}} n_{\text{train}} T)$, and the time complexity of the test using QNNs

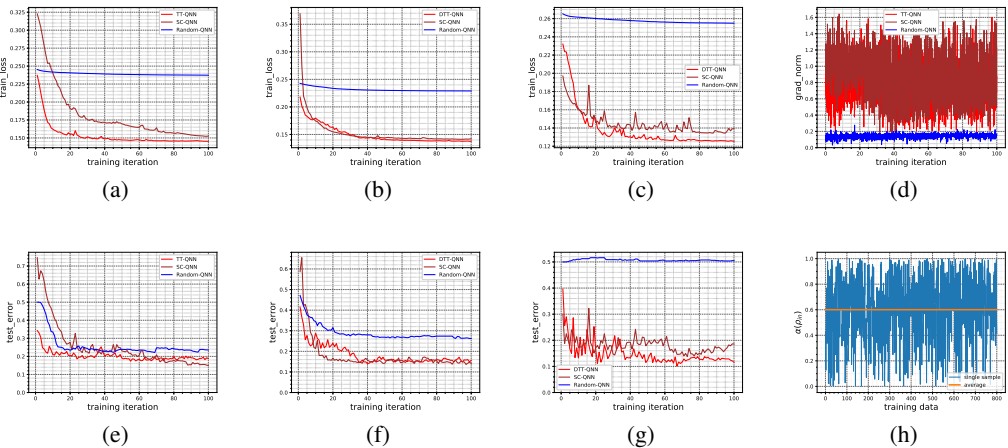

Figure 5: Simulations on the MNIST binary classification between $(0, 2)$. The training loss and the test error during the training iteration are illustrated in Figures 5(a), 5(e) for the n=8 case, Figures 5(b), 5(f) for the n=10 case, and Figures 5(c), 5(g) for the n=12 case. The gradient norm of objective functions and the term $\alpha(\rho_{\text{in}})$ during the training are shown in Figures 5(d) and 5(h), respectively for the n=8 case.

is $\mathcal{O}(n_{\text{gate}} n_{\text{test}})$. We emphasize that directly comparing the time complexity of QNNs with classical NNs is unfair due to different parameter strategies. However, the complexity of QNNs indeed shows a polylogarithmic dependence on the dimension of the input data if the number of gates and parameters are polynomial to the number of qubits. Specifically, both the TT-QNN and the SC-QNN equipped with the $L$-layer encoding circuit in this work have $\mathcal{O}(nL)$ gates and $\mathcal{O}(n)$ parameters, which lead to the training complexity $\mathcal{O}(n_{\text{train}} n^2 LT)$ and the test complexity $\mathcal{O}(n_{\text{test}} nL)$.

## 4.2 NUMERICAL SIMULATIONS

To analyze the practical performance of QNNs for binary classification tasks, we simulate the training and test of QNNs on the MNIST handwritten dataset. The $28 \times 28$ size image is sampled into $16 \times 16$, $32 \times 32$, and $64 \times 64$ to fit QNNs for qubit number $n \in \{8, 10, 12\}$. We set the parameter in SC-QNNs as $n_c = 4$ for all qubit number settings. Note that the based on the tree structure, the qubit number of original TT-QNNs is limited to the power of 2. To analysis the behavior of TT-QNNs for general qubit numbers, we modify TT-QNNs into Deformed Tree Tensor (DTT) QNNs. The gradient norm for DTT-QNNs is lower bounded by $\mathcal{O}(1/n)$, which has a similar form of that for TT-QNNs. We provide more details of DTT-QNNs in Appendix F and denote DTT-QNNs as TT-QNNs in simulation parts (Section 4.2 and Appendix A) for the convenience.

We construct the encoding circuits in Section 3.3 with the number of alternating layers $L = 1$ for 400 training samples and 400 test samples in each class. The TT-QNN and the SC-QNN is compared to the QNN with the random structure. To make a fair comparison, we set the numbers of RY and CNOT gates in the random QNN to be the same with the TT-QNN and the SC-QNN. The objective function of the random QNN is defined as the average of the expectation of the observable $\sigma_3$ for all qubits in the circuit. The number of the training iteration is 100, the batch size is 20, and the decayed learning rate is adopted as $\{1.00, 0.75, 0.50, 0.25\}$. We set $n_{\text{train}} = 200$ and $n_{\text{test}} = 1000$ as numbers of measurements for estimating quantum observables during the training and test stages, respectively. All experiments are simulated through the PennyLane Python package (Bergholm et al., 2020).

Firstly, we explain our results about QNNs with different qubit numbers in Figure 5. We train TT-QNNs, SC-QNNs, and Random-QNNs with the stochastic gradient descent method described in Algorithm 2 for images in the class $(0, 2)$ and the qubit number $n \in \{8, 10, 12\}$. The total loss is defined as the average of the single-input loss. The training loss and the test error during the training iteration are illustrated in Figures 5(a), 5(e) for the n=8 case, Figures 5(b), 5(f) for the

| Class pair | TT-QNN test (F1-0, F1-1) | SC-QNN test (F1-0, F1-1) | Random-QNN test (F1-0, F1-1) |
|---|---|---|---|
| 0-1 | 0.959 (0.958, 0.960) | **0.979** (0.979, 0.979) | 0.920 (0.915 , 0.925) |
| 0-2 | 0.844 (0.852, 0.834) | **0.859** (0.859, 0.858) | 0.738 (0.751, 0.722) |
| 0-3 | 0.835 (0.810, 0.854) | **0.859** (0.857, 0.860) | 0.659 (0.680, 0.635) |
| 0-4 | **0.938** (0.938, 0.938) | 0.925 (0.925, 0.925) | 0.779 (0.796, 0.759) |
| 1-2 | 0.929 (0.930, 0.927) | **0.965** (0.965, 0.965) | 0.700 (0.728, 0.666) |
| 1-3 | **0.938** (0.941, 0.934) | 0.881 (0.879, 0.884) | 0.765 (0.773, 0.756) |
| 1-4 | 0.821 (0.837, 0.802) | **0.871** (0.879, 0.862) | 0.705 (0.737, 0.664) |
| 2-3 | 0.814 (0.791, 0.832) | **0.820** (0.810, 0.829) | 0.709 (0.743, 0.664) |
| 2-4 | 0.931 (0.928, 0.934) | **0.935** (0.933, 0.937) | 0.645 (0.728, 0.487) |
| 3-4 | 0.931 (0.930, 0.933) | **0.944** (0.943, 0.945) | 0.819 (0.839, 0.793) |

Table 1: The test accuracy and F1-scores for different class pairs (qubit number = 10).

n=10 case, and Figures 5(c), 5(g) for the n=12 case. The test error of the TT-QNN, the SC-QNN and the Random-QNN converge to around 0.2 for the n=8 case. As the qubit number increases, the converged test error of both TT-QNNs and SC-QNNs remains lower than 0.2, while that of Random-QNNs increases to 0.26 and 0.50 for n=10 and n=12 case, respectively. The training loss of both TT-QNNs and SC-QNNs converges to around 0.15 for all qubit number settings, while that of Random-QNNs remains higher than 0.22. Both the training loss and the test error results show that TT-QNNs and SC-QNNs have better trainability and accuracy on the binary classification compared with Random-QNNs. We record the $l_2$-norm of the gradient during the training for the n=8 case in Figure 5(d). The gradient norm for the TT-QNN and the SC-QNN is mostly distributed in $[0.4, 1.4]$, which is significantly larger than the gradient norm for the Random-QNN that is mostly distributed in $[0.1, 0.2]$. As shown in Figure 5(d), the gradient norm verifies the lower bounds in the Theorem 3.1. Moreover, we calculate the term $\alpha(\rho_{\text{in}})$ defined in Theorem 3.1 and show the result in Figure 5(h). The average of $\alpha(\rho_{\text{in}})$ is around 0.6, which is lower bounded by the theoretical result $\frac{1}{4}$ in Theorem 3.2 ($L = 1$).

Secondly, we explain our results about QNNs on the binary classification with different class pairs. We conduct the binary classification with the same super parameters mentioned before for 10-qubit QNNs, and the test accuracy and F1-scores for all class pairs $\{i, j\} \subset \{0, 1, 2, 3, 4\}$ are provided in Table 1. The F1-0 denotes the F1-score when treats the former class to be positive, and the F1-1 denotes the F1-score for the other case. As shown in Table 1, TT-QNNs and SC-QNNs have higher test accuracy and F1-score than Random-QNNs for all class pairs in Table 1. Specifically, test accuracy of TT-QNNs and SC-QNNs exceed that of Random-QNNs by more than $10\%$ for all class pairs except the $(0, 1)$ which is relatively easy to classify.

In conclusion, both TT-QNNs and SC-QNNs show better trainability and accuracy on binary classification tasks compared with the random structure QNN, and all theorems are verified by experiments. We provide more experimental details and results about the input model and other classification tasks in Appendix A.

## 5 CONCLUSIONS

In this work, we analyze the vanishing gradient problem in quantum neural networks. We prove that the gradient norm of $n$-qubit quantum neural networks with the tree tensor structure and the step controlled structure are lower bounded by $\Omega(\frac{1}{n})$ and $\Omega((\frac{1}{2})^{n_c})$, respectively. The bound guarantees the trainability of TT-QNNs and SC-QNNs on related machine learning tasks. Our theoretical framework requires fewer assumptions than previous works and meets constraints on quantum neural networks for near-term quantum computers. Compared with the random structure QNN which is known to be suffered from the barren plateaus problem, both TT-QNNs and SC-QNNs show better trainability and accuracy on the binary classification task. We hope the paper could inspire future works on the trainability of QNNs with different architectures and other quantum machine learning algorithms.

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

## A    NUMERICAL SIMULATIONS

In this section, we provide more experimental details about the input model and other binary classification tasks.

### A.1    QUANTUM INPUT MODEL FOR THE MNIST DATASET

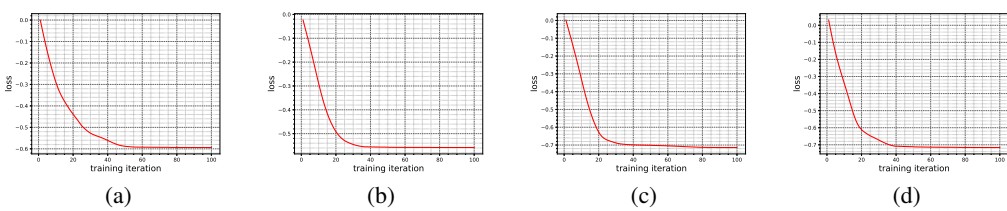

Figure 6: The training loss of the input model for MNIST images in class $\{0, 1, 2, 3\}$.

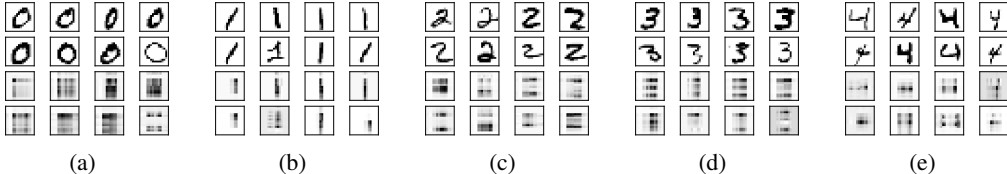

Figure 7: The visualization of the encoding circuit for MNIST images in class $\{0, 1, 2, 3, 4\}$.

In this section, we discuss the training of the input model (Algorithm 1) in detail. We construct the encoding circuits in Section 3.3 for each training or test data with the number of alternating layers $L = 1$. The number of the training iteration is 100. We adopt the decayed learning rate as $\{0.100, 0.075, 0.050, 0.025\}$. We illustrate the loss function defined in Eq. 7 during the training of Algorithm 1 for label in $\{0, 1, 2, 3\}$ for the n=8 case in Figure 6, in which we show the training of the input model for one image per sub-figure. All shown loss functions converge to around -0.6 after 60 iterations.

For a better understanding to the input model, we provide the visualization of the encoding circuit in Figure 7. We notice that the encoding circuit could only catch a few features from the input data (except the image 1 which shows good results). Despite this, we obtain relatively good results on binary classification tasks which employ the mentioned encoding circuit.

Apart from the binary classification tasks using QNNs equipped with the encoding circuit provided in Section 3.3, we perform some experiments such that the encoding circuit is replaced by the exact amplitude encoding, which is commonly used in existing quantum machine learning algorithms. Figure 8 demonstrates the simulation on MNIST classifications between images $(0, 2)$, which shows the convergence of training loss (Figure 8(a)) and the test error (Figure 8(b)). The norm of the gradient is counted in Figure 8(c), in which both the TT-QNN and the SC-QNN show larger gradient norm than the Random-QNN. Thus, the trainability of TT-QNNs and SC-QNNs remains when replace the encoding circuit with the exact amplitude encoding.

The training and test accuracy for other class pairs are summarized in Table 2. We notice that compared with QNNs using the encoding circuit, QNNs with the exact encoding tend to have better

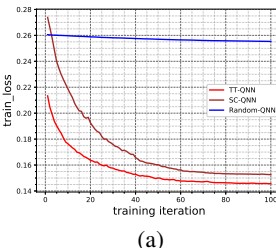 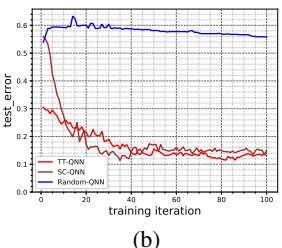 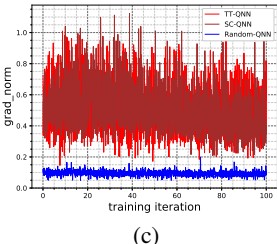

(a)             (b)             (c)

Figure 8: The training of binary classification task on the MNIST image bettwen classes $(0, 2)$ using 8-qubit QNNs with the exact amplitude encoding. The training loss and the test error during the training iteration are illustrated in Figures 8(a), 8(b), respectively. The gradient norm of objective functions during the training is shown in Figures 8(c).

| Class pair | TT-QNN
training, test | TT-QNN (exact encoding)
training, test | SC-QNN
training, test | SC-QNN (exact encoding)
training, test |
|---|---|---|---|---|
| 0-1 | 0.959, 0.981 | 0.980, **0.988** | 0.958, 0.969 | 0.986, 0.988 |
| 0-2 | 0.850, 0.812 | 0.902, **0.869** | 0.840, 0.850 | 0.881, 0.850 |
| 1-2 | 0.896, 0.909 | 0.924, 0.925 | 0.932, **0.943** | 0.919, 0.910 |

Table 2: The training and test accuracy of TT-QNNs and SC-QNNs for different encoding strategies and different class pairs.

performance on the accuracy, which is reasonable since the exact amplitude encoding remains all information of the input data.

## A.2 QNNs WITH DIFFERENT QUBIT SIZES

We summarize the results of training and test accuracy for different qubit numbers in Table 3, which corresponds to the main results presented in Figure 5 in Section 4.2. As shown in Table 3, both the training and test accuracy of TT-QNNs and SC-QNNs remain at a high level for all qubit number settings, while the training and test accuracy of Random-QNNs decrease to around 0.5 for the 12-qubit case, which means the Random-QNN cannot classify better than a random guessing.

| Qubit number | TT-QNN
training, test | SC-QNN
training, test | Random-QNN
training, test |
|---|---|---|---|
| 8 | 0.850, 0.812 | 0.840, **0.850** | 0.741, 0.765 |
| 10 | 0.865, 0.844 | 0.871, **0.859** | 0.804, 0.738 |
| 12 | 0.884, **0.884** | 0.844, 0.814 | 0.497, 0.495 |

Table 3: The training and test accuracy for different qubit numbers $\{8, 10, 12\}$ on the classification between $(0, 2)$ classes.

## A.3 QNNs WITH DIFFERENT LABEL PAIRS

We summarize the results of training and test accuracy, along with the F1-score, for QNNs on the classification between different label pairs in Table 4 and Table 5, for qubit number 8 and 10, respectively. For all label pairs, TT-QNNs and SC-QNNs show higher performance of the training accuracy, the test accuracy, and the F1-scores than that of Random-QNNs. Moreover, most of test accuracy of Random-QNNs drop for the same class pair when the qubit number is increased from 8 to 10, which suggest the trainability of Random-QNNs get worse as the qubit number increases.

| Class pair | TT-QNN training, test (F1-0, F1-1) | SC-QNN training, test (F1-0, F1-1) | Random-QNN training, test (F1-0, F1-1) |
|---|---|---|---|
| 0-1 | 0.959, **0.981** (0.981, 0.981) | 0.958, 0.969 (0.969, 0.969) | 0.954, 0.927 (0.923, 0.932) |
| 0-2 | 0.850, 0.812 (0.820, 0.804) | 0.840, **0.850** (0.847, 0.853) | 0.741, 0.765 (0.790, 0.734) |
| 0-3 | 0.871, 0.856 (0.845, 0.866) | 0.870, **0.861** (0.853, 0.869) | 0.722, 0.728 (0.734, 0.721) |
| 0-4 | 0.919, 0.894 (0.891, 0.896) | 0.934, **0.910** (0.907, 0.913) | 0.831, 0.766 (0.773, 0.759) |
| 1-2 | 0.896, 0.909 (0.914, 0.903) | 0.932, **0.943** (0.943, 0.942) | 0.627, 0.601 (0.704, 0.390) |
| 1-3 | 0.940, **0.940** (0.943, 0.937) | 0.922, 0.905 (0.912, 0.896) | 0.802, 0.794 (0.826, 0.747) |
| 1-4 | 0.891, **0.938** (0.938, 0.937) | 0.885, 0.915 (0.917, 0.912) | 0.850, 0.871 (0.881, 0.860) |
| 2-3 | 0.805, 0.771 (0.750, 0.789) | 0.824, **0.790** (0.761, 0.813) | 0.684, 0.666 (0.694, 0.633) |
| 2-4 | 0.810, 0.871 (0.873, 0.869) | 0.845, **0.901** (0.899, 0.904) | 0.738, 0.733 (0.752, 0.710) |
| 3-4 | 0.920, **0.926** (0.922, 0.930) | 0.939, 0.901 (0.898, 0.904) | 0.695, 0.701 (0.751, 0.627) |

Table 4: The classification accuracy for different class pairs (n=8).

| Class pair | TT-QNN training, test (F1-0, F1-1) | SC-QNN training, test (F1-0, F1-1) | Random-QNN training, test (F1-0, F1-1) |
|---|---|---|---|
| 0-1 | 0.965, 0.959 (0.958, 0.960) | 0.959, **0.979** (0.979, 0.979) | 0.922, 0.920 (0.915 , 0.925) |
| 0-2 | 0.865, 0.844 (0.852, 0.834) | 0.871, **0.859** (0.859, 0.858) | 0.804, 0.738 (0.751, 0.722) |
| 0-3 | 0.870, 0.835 (0.810, 0.854) | 0.871, **0.859** (0.857, 0.860) | 0.654, 0.659 (0.680, 0.635) |
| 0-4 | 0.945, **0.938** (0.938, 0.938) | 0.938, 0.925 (0.925, 0.925) | 0.839, 0.779 (0.796, 0.759) |
| 1-2 | 0.915, 0.929 (0.930, 0.927) | 0.930, **0.965** (0.965, 0.965) | 0.693, 0.700 (0.728, 0.666) |
| 1-3 | 0.944, **0.938** (0.941, 0.934) | 0.899, 0.881 (0.879, 0.884) | 0.759, 0.765 (0.773, 0.756) |
| 1-4 | 0.844, 0.821 (0.837, 0.802) | 0.854, **0.871** (0.879, 0.862) | 0.726, 0.705 (0.737, 0.664) |
| 2-3 | 0.829, 0.814 (0.791, 0.832) | 0.855, **0.820** (0.810, 0.829) | 0.754, 0.709 (0.743, 0.664) |
| 2-4 | 0.875, 0.931 (0.928, 0.934) | 0.889, **0.935** (0.933, 0.937) | 0.649, 0.645 (0.728, 0.487) |
| 3-4 | 0.940, 0.931 (0.930, 0.933) | 0.944, **0.944** (0.943, 0.945) | 0.876, 0.819 (0.839, 0.793) |

Table 5: The classification accuracy for different class pairs (n=10).

## A.4 QNNs with different rotation gates

In this section, we simulate variants of TT-QNNs and SC-QNNs such that single-qubit gate operations are extended from {RY} to {RX, RY, RZ}. Results on the binary classification between MNIST image $(0, 2)$ using 8-qubit QNNs are provided in Figure 9. The training loss converges to around 0.23 and 0.175 for the TT-QNN and the SC-QNN, respectively, and the test error converges to around 0.3 for both the TT-QNN and the SC-QNN. We remark that based on results in Figures 5(a) and Figure 5(e), the training loss of original TT-QNN and SC-QNN converge at around 0.15, and the test error converge at around 0.20 and 0.15, respectively. Thus, both the TT-QNN and the SC-QNN show the worse performance than original QNNs when employing the extended gate set {RX, RY, RZ}. Another result is provided in Table 6 which shows the difference on the training and test accuracy.

As a conclusion, employing gate set {RX, RY, RZ} could worse the performance of the QNNs on real-world problems, which may due to the fact that real-world data lie in the real space, while operations {RX, RZ} introduce the imaginary term to the state.

| Class pair | TT-QNN ({RY}) training, test | TT-QNN ({RX, RY, RZ}) training, test | SC-QNN ({RY}) training, test | SC-QNN ({RX, RY, RZ}) training, test |
|---|---|---|---|---|
| 0-2 | 0.850, 0.812 | 0.656, 0.699 | 0.840, 0.850 | 0.785, 0.731 |

Table 6: The training and test accuracy of TT-QNNs and SC-QNNs for different single-qubit gate settings.

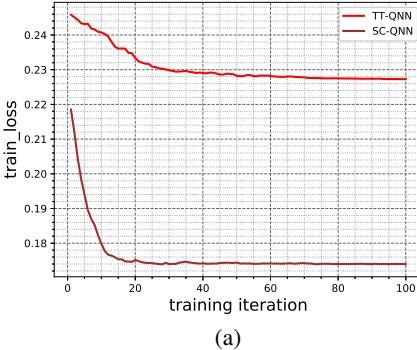 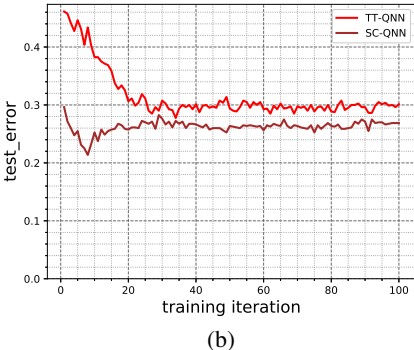

(a)                    (b)

Figure 9: The training of binary classification task on the MNIST image bettwen classes $(0, 2)$ using 8-qubit QNNs with the extended gate operation {RX, RY, RZ}. The training loss and the test error during the training iteration are illustrated in Figures 9(a), 9(b), respectively.

## B   NOTES ABOUT THE UNITARY 2-DESIGN

In this section, we introduce the notion of the unitary 2-design. Consider the finite gate set $S = \{G_i\}_{i=1}^{|S|}$ in the $d$-dimensional Hilbert space. We denote $U(d)$ as the unitary gate group with the dimension $d$. We denote $P_{t,t}(G)$ as the polynomial function which has the degree at most $t$ on the matrix elements of $G$ and at most $t$ on the matrix elements of $G^\dagger$. Then, we could say the set $S$ to be the unitary $t$-design if and only if for every function $P_{t,t}(\cdot)$, Eq. (9) holds:

$$\frac{1}{|S|} \sum_{G \in S} P_{t,t}(G) = \int_{U(d)} d\mu(G) P_{t,t}(G), \tag{9}$$

where $d\mu(\cdot)$ denotes the Haar distribution. The Haar distribution $d\mu(\cdot)$ is defined that for any function $f$ and any matrix $K \in U(d)$,

$$\int_{U(d)} d\mu(G) f(G) = \int_{U(d)} d\mu(G) f(KG) = \int_{U(d)} d\mu(G) f(GK).$$

The form in the right side of (9) can be viewed as the average or the expectation of the function $P_{t,t}(G)$. We remark that only the parameterized gates $RY = e^{-i\theta\sigma_2}$ could not form a universal gate set even in the single-qubit space $U(2)$, thus quantum circuits employing parameterized $RY$ gates could not form the 2-design. This is only a simple introduction about the unitary 2-design, and we refer readers to Puchała & Miszczak (2011) and Cerezo et al. (2020) for more detail.

## C   TECHNICAL LEMMAS

In this section we provide some technical lemmas.

**Lemma C.1.** *Let CNOT* $= \sigma_0 \otimes |0\rangle\langle 0| + \sigma_1 \otimes |1\rangle\langle 1|$. *Then*

$$CNOT(\sigma_j \otimes \sigma_k)CNOT^\dagger = (\delta_{j0} + \delta_{j1})(\delta_{k0} + \delta_{k3})\sigma_j \otimes \sigma_k + (\delta_{j0} + \delta_{j1})(\delta_{k1} + \delta_{k2})\sigma_j \sigma_1 \otimes \sigma_k$$
$$+ (\delta_{j2} + \delta_{j3})(\delta_{k0} + \delta_{k3})\sigma_j \otimes \sigma_k \sigma_3 - (\delta_{j2} + \delta_{j3})(\delta_{k1} + \delta_{k2})\sigma_j \sigma_1 \otimes \sigma_k \sigma_3.$$

*Further for the case* $\sigma_k = \sigma_0$,

$$CNOT(\sigma_j \otimes \sigma_0)CNOT^\dagger = (\delta_{j0} + \delta_{j1})\sigma_j \otimes \sigma_0 + (\delta_{j2} + \delta_{j3})\sigma_j \otimes \sigma_3.$$

*Proof.*

$$CNOT(\sigma_j \otimes \sigma_k)CNOT^\dagger$$
$$= (\sigma_0 \otimes |0\rangle\langle 0| + \sigma_1 \otimes |1\rangle\langle 1|)(\sigma_j \otimes \sigma_k)(\sigma_0 \otimes |0\rangle\langle 0| + \sigma_1 \otimes |1\rangle\langle 1|)$$

$$
\begin{aligned}
=& \left( \sigma_0 \otimes \frac{\sigma_0 + \sigma_3}{2} + \sigma_1 \otimes \frac{\sigma_0 - \sigma_3}{2} \right) (\sigma_j \otimes \sigma_k) \left( \sigma_0 \otimes \frac{\sigma_0 + \sigma_3}{2} + \sigma_1 \otimes \frac{\sigma_0 - \sigma_3}{2} \right) \\
=& \frac{1}{4} \left( \sigma_j \otimes \sigma_k + \sigma_1 \sigma_j \sigma_1 \otimes \sigma_k + \sigma_j \otimes \sigma_3 \sigma_k \sigma_3 + \sigma_1 \sigma_j \sigma_1 \otimes \sigma_3 \sigma_k \sigma_3 \right) \\
& + \frac{1}{4} \left( \sigma_j \sigma_1 \otimes \sigma_k + \sigma_1 \sigma_j \otimes \sigma_k - \sigma_j \sigma_1 \otimes \sigma_3 \sigma_k \sigma_3 - \sigma_1 \sigma_j \otimes \sigma_3 \sigma_k \sigma_3 \right) \\
& + \frac{1}{4} \left( \sigma_j \otimes \sigma_k \sigma_3 + \sigma_j \otimes \sigma_3 \sigma_k - \sigma_1 \sigma_j \sigma_1 \otimes \sigma_k \sigma_3 - \sigma_1 \sigma_j \sigma_1 \otimes \sigma_3 \sigma_k \right) \\
& + \frac{1}{4} \left( \sigma_j \sigma_1 \otimes \sigma_3 \sigma_k - \sigma_j \sigma_1 \otimes \sigma_k \sigma_3 + \sigma_1 \sigma_j \otimes \sigma_k \sigma_3 - \sigma_1 \sigma_j \otimes \sigma_3 \sigma_k \right) \\
=& (\delta_{j0} + \delta_{j1})(\delta_{k0} + \delta_{k3}) \sigma_j \otimes \sigma_k + (\delta_{j0} + \delta_{j1})(\delta_{k1} + \delta_{k2}) \sigma_j \sigma_1 \otimes \sigma_k \\
& + (\delta_{j2} + \delta_{j3})(\delta_{k0} + \delta_{k3}) \sigma_j \otimes \sigma_k \sigma_3 - (\delta_{j2} + \delta_{j3})(\delta_{k1} + \delta_{k2}) \sigma_j \sigma_1 \otimes \sigma_k \sigma_3.
\end{aligned}
$$

For the case $\sigma_k = \sigma_0$, we have,
$$
\text{CNOT}(\sigma_j \otimes \sigma_0)\text{CNOT}^\dagger = (\delta_{j0} + \delta_{j1})\sigma_j \otimes \sigma_0 + (\delta_{j2} + \delta_{j3})\sigma_j \otimes \sigma_3.
$$

$\square$

**Lemma C.2.** *Let* $CZ = \sigma_0 \otimes |0\rangle\langle 0| + \sigma_3 \otimes |1\rangle\langle 1|$. *Then*
$$
\begin{aligned}
CZ(\sigma_j \otimes \sigma_k)CZ^\dagger =& (\delta_{j0} + \delta_{j3})(\delta_{k0} + \delta_{k3})\sigma_j \otimes \sigma_k + (\delta_{j0} + \delta_{j3})(\delta_{k1} + \delta_{k2})\sigma_j \sigma_3 \otimes \sigma_k \\
& + (\delta_{j1} + \delta_{j2})(\delta_{k0} + \delta_{k3})\sigma_j \otimes \sigma_k \sigma_3 - (\delta_{j1} + \delta_{j2})(\delta_{k1} + \delta_{k2})\sigma_j \sigma_3 \otimes \sigma_k \sigma_3.
\end{aligned}
$$
*Further for the case* $\sigma_k = \sigma_0$,
$$
CZ(\sigma_j \otimes \sigma_0)CZ^\dagger = (\delta_{j0} + \delta_{j3})\sigma_j \otimes \sigma_0 + (\delta_{j1} + \delta_{j2})\sigma_j \otimes \sigma_3.
$$

*Proof.*

$$
\begin{aligned}
& CZ(\sigma_j \otimes \sigma_k)CZ^\dagger \\
=& \left( \sigma_0 \otimes |0\rangle\langle 0| + \sigma_3 \otimes |1\rangle\langle 1| \right) (\sigma_j \otimes \sigma_k) \left( \sigma_0 \otimes |0\rangle\langle 0| + \sigma_3 \otimes |1\rangle\langle 1| \right) \\
=& \left( \sigma_0 \otimes \frac{\sigma_0 + \sigma_3}{2} + \sigma_3 \otimes \frac{\sigma_0 - \sigma_3}{2} \right) (\sigma_j \otimes \sigma_k) \left( \sigma_0 \otimes \frac{\sigma_0 + \sigma_3}{2} + \sigma_3 \otimes \frac{\sigma_0 - \sigma_3}{2} \right) \\
=& \frac{1}{4} \left( \sigma_j \otimes \sigma_k + \sigma_3 \sigma_j \sigma_3 \otimes \sigma_k + \sigma_j \otimes \sigma_3 \sigma_k \sigma_3 + \sigma_3 \sigma_j \sigma_3 \otimes \sigma_3 \sigma_k \sigma_3 \right) \\
& + \frac{1}{4} \left( \sigma_j \sigma_3 \otimes \sigma_k + \sigma_3 \sigma_j \otimes \sigma_k - \sigma_j \sigma_3 \otimes \sigma_3 \sigma_k \sigma_3 - \sigma_3 \sigma_j \otimes \sigma_3 \sigma_k \sigma_3 \right) \\
& + \frac{1}{4} \left( \sigma_j \otimes \sigma_k \sigma_3 + \sigma_j \otimes \sigma_3 \sigma_k - \sigma_3 \sigma_j \sigma_3 \otimes \sigma_k \sigma_3 - \sigma_3 \sigma_j \sigma_3 \otimes \sigma_3 \sigma_k \right) \\
& + \frac{1}{4} \left( \sigma_j \sigma_3 \otimes \sigma_3 \sigma_k - \sigma_j \sigma_3 \otimes \sigma_k \sigma_3 + \sigma_3 \sigma_j \otimes \sigma_k \sigma_3 - \sigma_3 \sigma_j \otimes \sigma_3 \sigma_k \right) \\
=& (\delta_{j0} + \delta_{j3})(\delta_{k0} + \delta_{k3}) \sigma_j \otimes \sigma_k + (\delta_{j0} + \delta_{j3})(\delta_{k1} + \delta_{k2}) \sigma_j \sigma_3 \otimes \sigma_k \\
& + (\delta_{j1} + \delta_{j2})(\delta_{k0} + \delta_{k3}) \sigma_j \otimes \sigma_k \sigma_3 - (\delta_{j1} + \delta_{j2})(\delta_{k1} + \delta_{k2}) \sigma_j \sigma_3 \otimes \sigma_k \sigma_3.
\end{aligned}
$$

For the case $\sigma_k = \sigma_0$, we have,
$$
CZ(\sigma_j \otimes \sigma_0)CZ^\dagger = (\delta_{j0} + \delta_{j3})\sigma_j \otimes \sigma_0 + (\delta_{j1} + \delta_{j2})\sigma_j \otimes \sigma_3.
$$

$\square$

**Lemma C.3.** *Let* $\theta$ *be a variable with uniform distribution in* $[0, 2\pi]$. *Let* $A, C : \mathcal{H}_2 \to \mathcal{H}_2$ *be arbitrary linear operators and let* $B = D = \sigma_j$ *be arbitrary Pauli matrices, where* $j \in \{0, 1, 2, 3\}$. *Then*

$$
\mathbb{E}_\theta Tr[WAW^\dagger B]Tr[WCW^\dagger D] = \frac{1}{2\pi} \int_0^{2\pi} Tr[WAW^\dagger B]Tr[WCW^\dagger D]d\theta \tag{10}
$$

$$
= \left[ \frac{1}{2} + \frac{\delta_{j0} + \delta_{jk}}{2} \right] Tr[AB]Tr[CD] + \left[ -\frac{1}{2} + \frac{\delta_{j0} + \delta_{jk}}{2} \right] Tr[AB\sigma_k]Tr[CD\sigma_k], \tag{11}
$$

*where* $W = e^{-i\theta\sigma_k}$ *and* $k \in \{1, 2, 3\}$.

*Proof.* First we simply replace the term $W = e^{-i\theta\sigma_k} = I\cos\theta - i\sigma_k\sin\theta$.

$$\frac{1}{2\pi}\int_0^{2\pi} d\theta \mathrm{Tr}[WAW^\dagger B]\mathrm{Tr}[WCW^\dagger D]$$

$$=\frac{1}{2\pi}\int_0^{2\pi} d\theta \mathrm{Tr}[(I\cos\theta - i\sigma_k\sin\theta)A(I\cos\theta + i\sigma_k\sin\theta)B] \cdot \mathrm{Tr}[(I\cos\theta - i\sigma_k\sin\theta)C(I\cos\theta + i\sigma_k\sin\theta)D]$$

$$=\frac{1}{2\pi}\int_0^{2\pi} d\theta \left\{\cos^2\theta\mathrm{Tr}[AB] - i\sin\theta\cos\theta\mathrm{Tr}[\sigma_k AB] + i\sin\theta\cos\theta\mathrm{Tr}[A\sigma_k B] + \sin^2\theta\mathrm{Tr}[\sigma_k A\sigma_k B]\right\}$$

$$\cdot \left\{\cos^2\theta\mathrm{Tr}[CD] - i\sin\theta\cos\theta\mathrm{Tr}[\sigma_k CD] + i\sin\theta\cos\theta\mathrm{Tr}[C\sigma_k D] + \sin^2\theta\mathrm{Tr}[\sigma_k C\sigma_k D]\right\}.$$

We remark that:

$$\frac{1}{2\pi}\int_0^{2\pi} d\theta\cos^4\theta = \frac{3}{8}, \tag{12}$$

$$\frac{1}{2\pi}\int_0^{2\pi} d\theta\sin^4\theta = \frac{3}{8}, \tag{13}$$

$$\frac{1}{2\pi}\int_0^{2\pi} d\theta\cos^2\theta\sin^2\theta = \frac{1}{8}, \tag{14}$$

$$\frac{1}{2\pi}\int_0^{2\pi} d\theta\cos^3\theta\sin\theta = 0, \tag{15}$$

$$\frac{1}{2\pi}\int_0^{2\pi} d\theta\cos\theta\sin^3\theta = 0. \tag{16}$$

Then

$$\text{The integration term} = \frac{3}{8}\mathrm{Tr}[AB]\mathrm{Tr}[CD] + \frac{3}{8}\mathrm{Tr}[\sigma_k A\sigma_k B]\mathrm{Tr}[\sigma_k C\sigma_k D]$$

$$+ \frac{1}{8}\mathrm{Tr}[AB]\mathrm{Tr}[\sigma_k C\sigma_k D] + \frac{1}{8}\mathrm{Tr}[\sigma_k A\sigma_k B]\mathrm{Tr}[CD]$$

$$- \frac{1}{8}\mathrm{Tr}[\sigma_k AB]\mathrm{Tr}[\sigma_k CD] - \frac{1}{8}\mathrm{Tr}[A\sigma_k B]\mathrm{Tr}[C\sigma_k D]$$

$$+ \frac{1}{8}\mathrm{Tr}[\sigma_k AB]\mathrm{Tr}[C\sigma_k D] + \frac{1}{8}\mathrm{Tr}[A\sigma_k B]\mathrm{Tr}[\sigma_k CD]$$

$$= \mathrm{Tr}[AB]\mathrm{Tr}[CD]\left[\frac{1}{2} + \frac{\delta_{j0} + \delta_{jk}}{2}\right]$$

$$+ \mathrm{Tr}[AB\sigma_k]\mathrm{Tr}[CD\sigma_k]\left[-\frac{1}{2} + \frac{\delta_{j0} + \delta_{jk}}{2}\right].$$

The last equation is derived by noticing that for $B = \sigma_j$,

$$\mathrm{Tr}[\sigma_k A\sigma_k B] = \mathrm{Tr}[A\sigma_k\sigma_j\sigma_k]$$
$$= [2(\delta_{j0} + \delta_{jk}) - 1]\cdot\mathrm{Tr}[A\sigma_j]$$
$$= [2(\delta_{j0} + \delta_{jk}) - 1]\cdot\mathrm{Tr}[AB],$$
$$\mathrm{Tr}[\sigma_k AB] - \mathrm{Tr}[A\sigma_k B] = \mathrm{Tr}[A\sigma_j\sigma_k] - \mathrm{Tr}[A\sigma_k\sigma_j]$$
$$= 2(1 - \delta_{j0} - \delta_{jk})\cdot\mathrm{Tr}[A\sigma_j\sigma_k]$$
$$= 2(1 - \delta_{j0} - \delta_{jk})\cdot\mathrm{Tr}[AB\sigma_k],$$

while similar forms hold for $D = \sigma_j$,

$$\mathrm{Tr}[\sigma_k C\sigma_k D] = \mathrm{Tr}[C\sigma_k\sigma_j\sigma_k]$$
$$= [2(\delta_{j0} + \delta_{jk}) - 1]\cdot\mathrm{Tr}[C\sigma_j]$$
$$= [2(\delta_{j0} + \delta_{jk}) - 1]\cdot\mathrm{Tr}[CD],$$
$$\mathrm{Tr}[\sigma_k CD] - \mathrm{Tr}[C\sigma_k D] = \mathrm{Tr}[C\sigma_j\sigma_k] - \mathrm{Tr}[C\sigma_k\sigma_j]$$
$$= 2(1 - \delta_{j0} - \delta_{jk})\cdot\mathrm{Tr}[C\sigma_j\sigma_k]$$
$$= 2(1 - \delta_{j0} - \delta_{jk})\cdot\mathrm{Tr}[CD\sigma_k].$$

$\square$

**Lemma C.4.** *Let $\theta$ be a variable with uniform distribution in $[0, 2\pi]$. Let $A, C : \mathcal{H}_2 \to \mathcal{H}_2$ be arbitrary linear operators and let $B = D = \sigma_j$ be arbitrary Pauli matrices, where $j \in \{0, 1, 2, 3\}$. Then*

$$\mathbb{E}_\theta Tr[GAW^\dagger B]Tr[GCW^\dagger D] = \frac{1}{2\pi} \int_0^{2\pi} Tr[GAW^\dagger B]Tr[GCW^\dagger D]d\theta$$

$$= \left[\frac{1}{2} - \frac{\delta_{j0} + \delta_{jk}}{2}\right] Tr[AB]Tr[CD] + \left[-\frac{1}{2} - \frac{\delta_{j0} + \delta_{jk}}{2}\right] Tr[AB\sigma_k]Tr[CD\sigma_k],$$

*where $W = e^{-i\theta\sigma_k}$, $G = \frac{\partial W}{\partial\theta}$ and $k \in \{1, 2, 3\}$.*

*Proof.* First we simply replace the term $W = e^{-i\theta\sigma_k} = I\cos\theta - i\sigma_k\sin\theta$ and $G = \frac{\partial W}{\partial\theta} = -I\sin\theta - i\sigma_k\cos\theta$.

$$\frac{1}{2\pi} \int_0^{2\pi} d\theta \text{Tr}[GAW^\dagger B]\text{Tr}[GCW^\dagger D]$$

$$= \frac{1}{2\pi} \int_0^{2\pi} d\theta \text{Tr}[(-I\sin\theta - i\sigma_k\cos\theta)A(I\cos\theta + i\sigma_k\sin\theta)B]\text{Tr}[(-I\sin\theta - i\sigma_k\cos\theta)C(I\cos\theta + i\sigma_k\sin\theta)D]$$

$$= \frac{1}{2\pi} \int_0^{2\pi} d\theta \left\{-\sin\theta\cos\theta\text{Tr}[AB] - i\cos^2\theta\text{Tr}[\sigma_k AB] - i\sin^2\theta\text{Tr}[A\sigma_k B] + \cos\theta\sin\theta\text{Tr}[\sigma_k A\sigma_k B]\right\}$$

$$\cdot \left\{-\sin\theta\cos\theta\text{Tr}[CD] - i\cos^2\theta\text{Tr}[\sigma_k CD] - i\sin^2\theta\text{Tr}[C\sigma_k D] + \cos\theta\sin\theta\text{Tr}[\sigma_k C\sigma_k D]\right\}.$$

The integration above could be simplified using equations 12-16,

$$\begin{aligned}
\text{The integration term} = &\frac{1}{8}\text{Tr}[AB]\text{Tr}[CD] + \frac{1}{8}\text{Tr}[\sigma_k A\sigma_k B]\text{Tr}[\sigma_k C\sigma_k D] \\
&- \frac{1}{8}\text{Tr}[AB]\text{Tr}[\sigma_k C\sigma_k D] - \frac{1}{8}\text{Tr}[\sigma_k A\sigma_k B]\text{Tr}[CD] \\
&- \frac{3}{8}\text{Tr}[\sigma_k AB]\text{Tr}[\sigma_k CD] - \frac{3}{8}\text{Tr}[A\sigma_k B]\text{Tr}[C\sigma_k D] \\
&- \frac{1}{8}\text{Tr}[\sigma_k AB]\text{Tr}[C\sigma_k D] - \frac{1}{8}\text{Tr}[A\sigma_k B]\text{Tr}[\sigma_k CD] \\
= &\text{Tr}[AB]\text{Tr}[CD]\left[\frac{1}{2} - \frac{\delta_{j0} + \delta_{jk}}{2}\right] \\
&+ \text{Tr}[AB\sigma_k]\text{Tr}[CD\sigma_k]\left[-\frac{1}{2} - \frac{\delta_{j0} + \delta_{jk}}{2}\right].
\end{aligned}$$

The last equation is derived by noticing that for $B = \sigma_j$,

$$\begin{aligned}
\text{Tr}[\sigma_k A\sigma_k B] &= \text{Tr}[A\sigma_k\sigma_j\sigma_k] \\
&= [2(\delta_{j0} + \delta_{jk}) - 1]\cdot\text{Tr}[A\sigma_j] \\
&= [2(\delta_{j0} + \delta_{jk}) - 1]\cdot\text{Tr}[AB], \\
\text{Tr}[\sigma_k AB] + \text{Tr}[A\sigma_k B] &= \text{Tr}[A\sigma_j\sigma_k] + \text{Tr}[A\sigma_k\sigma_j] \\
&= 2(\delta_{j0} + \delta_{jk})\cdot\text{Tr}[A\sigma_j\sigma_k] \\
&= 2(\delta_{j0} + \delta_{jk})\cdot\text{Tr}[AB\sigma_k],
\end{aligned}$$

while similar forms hold for $D = \sigma_j$,

$$\begin{aligned}
\text{Tr}[\sigma_k C\sigma_k D] &= \text{Tr}[C\sigma_k\sigma_j\sigma_k] \\
&= [2(\delta_{j0} + \delta_{jk}) - 1]\cdot\text{Tr}[C\sigma_j] \\
&= [2(\delta_{j0} + \delta_{jk}) - 1]\cdot\text{Tr}[CD], \\
\text{Tr}[\sigma_k CD] + \text{Tr}[C\sigma_k D] &= \text{Tr}[C\sigma_j\sigma_k] - \text{Tr}[C\sigma_k\sigma_j] \\
&= 2(\delta_{j0} + \delta_{jk})\cdot\text{Tr}[C\sigma_j\sigma_k] \\
&= 2(\delta_{j0} + \delta_{jk})\cdot\text{Tr}[CD\sigma_k].
\end{aligned}$$

$\square$

# D   THE PROOF OF THEOREM 3.1: THE TT PART

Now we begin the proof of Theorem 3.1.

*Proof.* Firstly we remark that by Lemma D.1, each partial derivative is calculated as

$$\frac{\partial f_{\text{TT}}}{\partial \theta_j^{(k)}} = \frac{1}{2} \left( \text{Tr}[O \cdot V_{\text{TT}}(\boldsymbol{\theta}_+)\rho_{\text{in}} V_{\text{TT}}(\boldsymbol{\theta}_+)^\dagger] - \text{Tr}[O \cdot V_{\text{TT}}(\boldsymbol{\theta}_-)\rho_{\text{in}} V_{\text{TT}}(\boldsymbol{\theta}_-)^\dagger] \right),$$

since the expectation of the quantum observable is bounded by $[-1, 1]$, the square of the partial derivative could be easily bounded as:

$$\left( \frac{\partial f_{\text{TT}}}{\partial \theta_j^{(k)}} \right)^2 \leq 1.$$

By summing up $2n - 1$ parameters, we obtain

$$\|\nabla_{\boldsymbol{\theta}} f_{\text{TT}}\|^2 = \sum_{j,k} \left( \frac{\partial f_{\text{TT}}}{\partial \theta_j^{(k)}} \right)^2 \leq 2n - 1.$$

On the other side, the lower bound could be derived as follows,

$$\mathbb{E}_{\boldsymbol{\theta}} \|\nabla_{\boldsymbol{\theta}} f_{\text{TT}}\|^2 \geq \sum_{j=1}^{1+\log n} \mathbb{E}_{\boldsymbol{\theta}} \left( \frac{\partial f_{\text{TT}}}{\partial \theta_j^{(1)}} \right)^2 \tag{17}$$

$$= \sum_{j=1}^{1+\log n} 4\mathbb{E}_{\boldsymbol{\theta}} \left( f_{\text{TT}} - \frac{1}{2} \right)^2 \tag{18}$$

$$\geq \frac{1 + \log n}{2n} \cdot \left( \text{Tr} \left[ \sigma_{(1,0,\cdots,0)} \cdot \rho_{\text{in}} \right]^2 + \text{Tr} \left[ \sigma_{(3,0,\cdots,0)} \cdot \rho_{\text{in}} \right]^2 \right), \tag{19}$$

where Eq. (18) is derived using Lemma D.2, and Eq. (19) is derived using Lemma D.3.

$\square$

Now we provide the detail and the proof of Lemmas D.1, D.2, D.3.

**Lemma D.1.** *Consider the objective function of the QNN defined as*

$$f(\boldsymbol{\theta}) = \frac{1 + \langle O \rangle}{2} = \frac{1 + Tr[O \cdot V(\boldsymbol{\theta})\rho_{in} V(\boldsymbol{\theta})^\dagger]}{2},$$

*where $\boldsymbol{\theta}$ encodes all parameters which participate the circuit as $e^{-i\theta_j \sigma_k}$, $k \in 1, 2, 3$, $\rho_{in}$ denotes the input state and $O$ is an arbitrary quantum observable. Then, the partial derivative of the function respect to the parameter $\theta_j$ could be calculated by*

$$\frac{\partial f}{\partial \theta_j} = \frac{1}{2} \left( Tr[O \cdot V(\boldsymbol{\theta}_+)\rho_{in} V(\boldsymbol{\theta}_+)^\dagger] - Tr[O \cdot V(\boldsymbol{\theta}_-)\rho_{in} V(\boldsymbol{\theta}_-)^\dagger] \right),$$

*where $\boldsymbol{\theta}_+ \equiv \boldsymbol{\theta} + \frac{\pi}{4}\boldsymbol{e}_j$ and $\boldsymbol{\theta}_- \equiv \boldsymbol{\theta} - \frac{\pi}{4}\boldsymbol{e}_j$.*

*Proof.* First we assume that the circuit $V(\boldsymbol{\theta})$ consists of $p$ parameters, and could be written in the sequence:

$$V(\boldsymbol{\theta}) = V_p(\theta_p) \cdot V_{p-1}(\theta_{p-1}) \cdots V_1(\theta_1),$$

such that each block $V_j$ contains only one parameter.

Consider the observable defined as $O' = V_{j+1}^\dagger \cdots V_p^\dagger O V_p \cdots V_{j+1}$ and the input state defined as $\rho'_{\text{in}} = V_{j-1} \cdots V_1 \rho_{\text{in}} V_1^\dagger \cdots V_{j-1}^\dagger$. The parameter shifting rule (Crooks, 2019) provides a gradient

calculation method for the single parameter case. For $f_j(\theta_j) = \text{Tr}[O' \cdot U(\theta_j) \rho'_{\text{in}} U(\theta_j)^\dagger]$, the gradient could be calculated as

$$\frac{df_j}{d\theta_j} = f_j(\theta_j + \frac{\pi}{4}) - f_j(\theta_j - \frac{\pi}{4}).$$

Thus, by inserting the form of $O'$ and $\rho'_{\text{in}}$, we could obtain

$$\frac{\partial f}{\partial \theta_j} = \frac{df_j}{d\theta_j} = f_j(\theta_j + \frac{\pi}{4}) - f_j(\theta_j - \frac{\pi}{4}) = \frac{1}{2}\left(\text{Tr}[O \cdot V(\boldsymbol{\theta}_+)\rho_{\text{in}}V(\boldsymbol{\theta}_+)^\dagger] - \text{Tr}[O \cdot V(\boldsymbol{\theta}_-)\rho_{\text{in}}V(\boldsymbol{\theta}_-)^\dagger]\right).$$

$$\square$$

**Lemma D.2.** *For the objective function $f_{TT}$ defined in Eq. (3), the following formula holds for every $j \in \{1, 2, \cdots, 1 + \log n\}$:*

$$\mathbb{E}_{\boldsymbol{\theta}}\left(\frac{\partial f_{TT}}{\partial \theta_j^{(1)}}\right)^2 = 4 \cdot \mathbb{E}_{\boldsymbol{\theta}}(f_{TT} - \frac{1}{2})^2, \tag{20}$$

*where the expectation is taken for all parameters in $\theta$ with uniform distribution in $[0, 2\pi]$.*

*Proof.* First we rewrite the formulation of $f_{TT}$ in detail:

$$f_{TT} = \frac{1}{2} + \frac{1}{2}\text{Tr}\left[\sigma_{(3,0,\cdots,0)} \cdot V_{m+1}CX_m \cdots CX_1V_1 \cdot \rho_{\text{in}} \cdot V_1^\dagger CX_1^\dagger \cdots CX_m^\dagger V_{m+1}^\dagger\right], \tag{21}$$

where $m = \log n$ and we denote

$$\sigma_{(i_1, i_2, \cdots, i_n)} \equiv \sigma_{i_1} \otimes \sigma_{i_2} \otimes \cdots \otimes \sigma_{i_n}.$$

The operation $V_\ell$ consists of $n \cdot 2^{1-\ell}$ single qubit rotations $W_\ell^{(j)} = e^{-i\sigma_2\theta_\ell^{(j)}}$ on the $(j-1) \cdot 2^{\ell-1}+1$-th qubit, where $j = 1, 2, \cdots, n \cdot 2^{1-\ell}$. The operation $CX_\ell$ consists of $n \cdot 2^{-\ell}$ CNOT gates, where each of them acts on the $(j-1) \cdot 2^\ell + 1$-th and $(j-0.5) \cdot 2^\ell + 1$-th qubit, for $j = 1, 2, \cdots, n \cdot 2^{-\ell}$.

Now we focus on the partial derivative of the function $f$ to the parameter $\theta_j^{(1)}$. We have:

$$\frac{\partial f_{TT}}{\partial \theta_j^{(1)}} = \frac{1}{2}\text{Tr}\left[\sigma_{(3,0,\cdots,0)} \cdot V_{m+1}CX_m \cdots \frac{\partial V_j}{\theta_j^{(1)}} \cdots CX_1V_1 \cdot \rho_{\text{in}} \cdot V_1^\dagger CX_1^\dagger \cdots V_j^\dagger \cdots CX_m^\dagger V_{m+1}^\dagger\right] \tag{22}$$

$$+ \frac{1}{2}\text{Tr}\left[\sigma_{(3,0,\cdots,0)} \cdot V_{m+1}CX_m \cdots V_j \cdots CX_1V_1 \cdot \rho_{\text{in}} \cdot V_1^\dagger CX_1^\dagger \cdots \frac{\partial V_j^\dagger}{\theta_j^{(1)}} \cdots CX_m^\dagger V_{m+1}^\dagger\right] \tag{23}$$

$$= \text{Tr}\left[\sigma_{(3,0,\cdots,0)} \cdot V_{m+1}CX_m \cdots \frac{\partial V_j}{\theta_j^{(1)}} \cdots CX_1V_1 \cdot \rho_{\text{in}} \cdot V_1^\dagger CX_1^\dagger \cdots V_j^\dagger \cdots CX_m^\dagger V_{m+1}^\dagger\right]. \tag{24}$$

The Eq. (24) holds because both terms in (22) and (23) except $\rho_{\text{in}}$ are real matrices, and $\rho_{\text{in}} = \rho_{\text{in}}^\dagger$.

The key idea to derive $\mathbb{E}_{\boldsymbol{\theta}}\left(\frac{\partial f_{TT}}{\partial \theta_j^{(1)}}\right)^2 = 4 \cdot \mathbb{E}_{\boldsymbol{\theta}}(f_{TT} - \frac{1}{2})^2$ is that for cases $B = D = \sigma_j \in \{\sigma_1, \sigma_3\}$, the term $\frac{\delta_{j0} + \delta_{j2}}{2} = 0$, which means Lemma C.3 and Lemma C.4 collapse to the same formulation:

$$\mathbb{E}_{\boldsymbol{\theta}}\text{Tr}[WAW^\dagger B]\text{Tr}[WCW^\dagger D] = \mathbb{E}_{\boldsymbol{\theta}}\text{Tr}[GAW^\dagger B]\text{Tr}[GCW^\dagger D]$$

$$= \frac{1}{2}\text{Tr}[A\sigma_1]\text{Tr}[C\sigma_1] + \frac{1}{2}\text{Tr}[A\sigma_3]\text{Tr}[C\sigma_3].$$

Now we write the analysis in detail.

$$\mathbb{E}_{\boldsymbol{\theta}}\left[\left(\frac{\partial f_{TT}}{\partial \theta_j^{(1)}}\right)^2 - 4(f_{TT} - \frac{1}{2})^2\right] \tag{25}$$

$$= \mathbb{E}_{\boldsymbol{\theta}_1} \cdots \mathbb{E}_{\boldsymbol{\theta}_m} \mathbb{E}_{\theta_{m+1}^{(1)}} \mathrm{Tr} \left[ \sigma_{(3,0,\cdots,0)} \cdot V_{m+1} CX_m A_m CX_m V_{m+1}^\dagger \right]^2 \tag{26}$$

$$- \mathbb{E}_{\boldsymbol{\theta}_1} \cdots \mathbb{E}_{\boldsymbol{\theta}_m} \mathbb{E}_{\theta_{m+1}^{(1)}} \mathrm{Tr} \left[ \sigma_{(3,0,\cdots,0)} \cdot V_{m+1} CX_m B_m CX_m V_{m+1}^\dagger \right]^2 \tag{27}$$

$$= \mathbb{E}_{\boldsymbol{\theta}_1} \cdots \mathbb{E}_{\boldsymbol{\theta}_m} \left\{ \frac{1}{2} \mathrm{Tr} \left[ \sigma_{(3,0,\cdots,0,3,0,\cdots,0)} \cdot A_m \right]^2 + \frac{1}{2} \mathrm{Tr} \left[ \sigma_{(1,0,\cdots,0,0,0,\cdots,0)} \cdot A_m \right]^2 \right\} \tag{28}$$

$$- \mathbb{E}_{\boldsymbol{\theta}_1} \cdots \mathbb{E}_{\boldsymbol{\theta}_m} \left\{ \frac{1}{2} \mathrm{Tr} \left[ \sigma_{(3,0,\cdots,0,3,0,\cdots,0)} \cdot B_m \right]^2 + \frac{1}{2} \mathrm{Tr} \left[ \sigma_{(1,0,\cdots,0,0,0,\cdots,0)} \cdot B_m \right]^2 \right\}, \tag{29}$$

where

$$A_m = V_m CX_{m-1} \cdots \frac{\partial V_j}{\theta_j^{(1)}} \cdots CX_1 V_1 \cdot \rho_{\mathrm{in}} \cdot V_1^\dagger CX_1^\dagger \cdots V_j^\dagger \cdots CX_{m-1}^\dagger V_m^\dagger,$$

$$B_m = V_m CX_{m-1} \cdots V_j \cdots CX_1 V_1 \cdot \rho_{\mathrm{in}} \cdot V_1^\dagger CX_1^\dagger \cdots V_j^\dagger \cdots CX_{m-1}^\dagger V_m^\dagger,$$

and Eq. (28,29) are derived using the collapsed form of Lemma C.1:

$$\mathrm{CNOT}(\sigma_1 \otimes \sigma_0)\mathrm{CNOT}^\dagger = \sigma_1 \otimes \sigma_0, \ \mathrm{CNOT}(\sigma_3 \otimes \sigma_0)\mathrm{CNOT}^\dagger = \sigma_3 \otimes \sigma_3,$$

and $\boldsymbol{\theta}_\ell$ denotes the vector consisted with all parameters in the $\ell$-th layer. The integration could be performed for parameters $\{\boldsymbol{\theta}_m, \boldsymbol{\theta}_{m-1}, \cdots, \boldsymbol{\theta}_{j+1}\}$. It is not hard to find that after the integration of the parameters $\boldsymbol{\theta}_{j+1}$, the term $\mathrm{Tr}[\sigma_{\boldsymbol{i}} \cdot A_j]^2$ and $\mathrm{Tr}[\sigma_{\boldsymbol{i}} \cdot B_j]^2$ have the opposite coefficients. Besides, the first index of each Pauli tensor product $\sigma_{(i_1,i_2,\cdots,i_n)}$ could only be $i_1 \in \{1, 3\}$ because of the Lemma C.3. So we could write

$$\mathbb{E}_{\boldsymbol{\theta}} \left[ \left( \frac{\partial f_{\mathrm{TT}}}{\partial \theta_j^{(1)}} \right)^2 - 4(f_{\mathrm{TT}} - \frac{1}{2})^2 \right] \tag{30}$$

$$= \mathbb{E}_{\boldsymbol{\theta}_1} \cdots \mathbb{E}_{\boldsymbol{\theta}_j} \left\{ \sum_{i_1 \in \{1,3\}} \sum_{i_2=0}^3 \cdots \sum_{i_n=0}^3 a_{\boldsymbol{i}} \mathrm{Tr} \left[ \sigma_{\boldsymbol{i}} \cdot A_j \right]^2 - a_{\boldsymbol{i}} \mathrm{Tr} \left[ \sigma_{\boldsymbol{i}} \cdot B_j \right]^2 \right\} \tag{31}$$

where

$$A_j = \frac{\partial V_j}{\partial \theta_j^{(1)}} CX_{j-1} \cdots CX_1 V_1 \cdot \rho_{\mathrm{in}} \cdot V_1^\dagger CX_1^\dagger \cdots CX_{j-1} V_j^\dagger$$

$$= (G_j^{(1)} \otimes I^{\otimes(n-1)}) A_j^{/\theta_j^{(1)}} (W_j^{(1)\dagger} \otimes I^{\otimes(n-1)}),$$

$$B_j = V_j CX_{j-1} \cdots CX_1 V_1 \cdot \rho_{\mathrm{in}} \cdot V_1^\dagger CX_1^\dagger \cdots CX_{j-1} V_j^\dagger$$

$$= (W_j^{(1)} \otimes I^{\otimes(n-1)}) A_j^{/\theta_j^{(1)}} (W_j^{(1)\dagger} \otimes I^{\otimes(n-1)}),$$

and $a_{\boldsymbol{i}}$ is the coefficient of the term $\mathrm{Tr}\left[\sigma_{\boldsymbol{i}} \cdot A_j\right]^2$. We denote $G_j^{(1)} = \frac{\partial W_j^{(1)}}{\partial \theta_j^{(1)}}$ and use $A_j^{/\theta_j^{(1)}}$ to denote the rest part of $A_j$ and $B_j$. By Lemma C.3 and Lemma C.4, we have

$$\mathbb{E}_{\theta_j^{(1)}} \left[ \mathrm{Tr} \left[ \sigma_{\boldsymbol{i}} \cdot A_j \right]^2 - \mathrm{Tr} \left[ \sigma_{\boldsymbol{i}} \cdot B_j \right]^2 \right] = 0,$$

since for the case $i_1 \in \{1, 3\}$, the term $\frac{\delta_{i_1 0} + \delta_{i_1 2}}{2} = 0$, which means Lemma C.3 and Lemma C.4 have the same formulation. Then, we derive the Eq. (20).

$\square$

**Lemma D.3.** *For the loss function $f$ defined in (3), we have:*

$$\mathbb{E}_{\boldsymbol{\theta}}(f_{\mathrm{TT}} - \frac{1}{2})^2 \geq \frac{\left\{ Tr \left[ \sigma_{(1,0,\cdots,0)} \cdot \rho_{in} \right] \right\}^2 + \left\{ Tr \left[ \sigma_{(3,0,\cdots,0)} \cdot \rho_{in} \right] \right\}^2}{8n}, \tag{32}$$

*where we denote*

$$\sigma_{(i_1,i_2,\cdots,i_n)} \equiv \sigma_{i_1} \otimes \sigma_{i_2} \otimes \cdots \otimes \sigma_{i_n},$$

*and the expectation is taken for all parameters in $\theta$ with uniform distributions in $[0, 2\pi]$.*

*Proof.* First we expand the function $f_{\text{TT}}$ in detail,

$$f_{\text{TT}} = \frac{1}{2} + \frac{1}{2}\text{Tr}\left[\sigma_{(3,0,\cdots,0)} \cdot V_{m+1}CX_m\cdots CX_1V_1 \cdot \rho_{\text{in}} \cdot V_1^\dagger CX_1^\dagger \cdots CX_m^\dagger V_{m+1}^\dagger\right], \quad (33)$$

where $m = \log n$. Now we consider the expectation of $(f_{\text{TT}} - \frac{1}{2})^2$ under the uniform distribution for $\theta_{m+1}^{(1)}$:

$$\mathbb{E}_{\theta_{m+1}^{(1)}}(f_{\text{TT}} - \frac{1}{2})^2 = \frac{1}{4}\mathbb{E}_{\theta_{m+1}^{(1)}}\left\{\text{Tr}\left[\sigma_{(3,0,\cdots,0)} \cdot V_{m+1}CX_m\cdots CX_1V_1 \cdot \rho_{\text{in}} \cdot V_1^\dagger CX_1^\dagger \cdots CX_m^\dagger V_{m+1}^\dagger\right]\right\}^2 \quad (34)$$

$$= \frac{1}{8}\left\{\text{Tr}\left[\sigma_{(3,0,\cdots,0)} \cdot A'\right]\right\}^2 + \frac{1}{8}\left\{\text{Tr}\left[\sigma_{(1,0,\cdots,0)} \cdot A'\right]\right\}^2 \quad (35)$$

$$= \frac{1}{8}\left\{\text{Tr}\left[\sigma_{(3,0,\cdots,0,3,0,\cdots,0)} \cdot A\right]\right\}^2 + \frac{1}{8}\left\{\text{Tr}\left[\sigma_{(1,0,\cdots,0,0,0,\cdots,0)} \cdot A\right]\right\}^2 \quad (36)$$

$$\geq \frac{1}{8}\left\{\text{Tr}\left[\sigma_{(1,0,\cdots,0,0,0,\cdots,0)} \cdot A\right]\right\}^2, \quad (37)$$

where

$$A' = CX_mV_m\cdots CX_1V_1 \cdot \rho_{\text{in}} \cdot V_1^\dagger CX_1^\dagger \cdots V_mCX_m^\dagger, \quad (38)$$

$$A = V_m\cdots CX_1V_1 \cdot \rho_{\text{in}} \cdot V_1^\dagger CX_1^\dagger \cdots V_m, \quad (39)$$

and Eq. (35) is derived using Lemma C.3, Eq. (36) is derived using the collapsed form of Lemma C.1:

$$\text{CNOT}(\sigma_1 \otimes \sigma_0)\text{CNOT}^\dagger = \sigma_1 \otimes \sigma_0, \ \text{CNOT}(\sigma_3 \otimes \sigma_0)\text{CNOT}^\dagger = \sigma_3 \otimes \sigma_3,$$

We remark that during the integration of the parameters $\{\theta_\ell^{(j)}\}$ in each layer $\ell \in \{1, 2, \cdots, m+1\}$, the coefficient of the term $\{\text{Tr}[\sigma_{(1,0,\cdots,0)} \cdot A]\}^2$ only times a factor $1/2$ for the case $j = 1$, and the coefficient remains for the cases $j > 1$ (check Lemma C.3 for detail). Since the formulation $\{\text{Tr}[\sigma_{(1,0,\cdots,0)} \cdot A]\}^2$ remains the same when merging the operation $CX_\ell$ with $\sigma_{(1,0,\cdots,0)}$, for $\ell \in \{1, 2, \cdots, m\}$, we could generate the following equation,

$$\mathbb{E}_{\boldsymbol{\theta}_2}\cdots\mathbb{E}_{\boldsymbol{\theta}_{m+1}}(f_{\text{TT}} - \frac{1}{2})^2 \geq (\frac{1}{2})^{m+2}\left\{\text{Tr}\left[\sigma_{(1,0,\cdots,0)} \cdot V_1 \cdot \rho_{\text{in}} \cdot V_1^\dagger\right]\right\}^2, \quad (40)$$

where $\boldsymbol{\theta}_\ell$ denotes the vector consisted with all parameters in the $\ell$-th layer.

Finally by using Lemma C.3, we could integrate the parameters $\{\theta_1^{(j)}\}_{j=1}^n$ in (40):

$$\mathbb{E}_{\boldsymbol{\theta}}(f_{\text{TT}} - \frac{1}{2})^2 = \mathbb{E}_{\boldsymbol{\theta}_1}\cdots\mathbb{E}_{\boldsymbol{\theta}_{m+1}}(f_{\text{TT}} - \frac{1}{2})^2 \quad (41)$$

$$\geq \frac{\left\{\text{Tr}\left[\sigma_{(1,0,\cdots,0)} \cdot \rho_{\text{in}}\right]\right\}^2 + \left\{\text{Tr}\left[\sigma_{(3,0,\cdots,0)} \cdot \rho_{\text{in}}\right]\right\}^2}{2^{m+3}} \quad (42)$$

$$= \frac{\left\{\text{Tr}\left[\sigma_{(1,0,\cdots,0)} \cdot \rho_{\text{in}}\right]\right\}^2 + \left\{\text{Tr}\left[\sigma_{(3,0,\cdots,0)} \cdot \rho_{\text{in}}\right]\right\}^2}{8n}. \quad (43)$$

$$\square$$

# E   THE QUANTUM INPUT MODEL

For the convenience of the analysis, we consider the encoding model that the number of alternating layers is $L$. The model begins with the inital state $|0\rangle^{\otimes n}$, where $n$ is the number of the qubit. Then we employ the $X$ gate to each qubit which transform the state into $|1\rangle^{\otimes n}$. Next we employ $L$ alternating layer operations, each of which contains four parts: a single qubit rotation layer denoted as $V_{2i-1}$, a $CZ$ gate layer denoted as $CZ_2$, again a single qubit rotation layer denoted as $V_{2i}$, and another $CZ$ gate layer with alternating structures denoted as $CZ_1$, for $i \in \{1, 2, \cdots, L\}$. Each single qubit gate contains a parameter encoded in the phase: $W_j^{(k)} = e^{-i\sigma_2\beta_j^{(k)}}$, and each single qubit rotation layer could be written as

$$V_j = V_j(\boldsymbol{\beta}_j) = W_j^{(1)} \otimes W_j^{(2)} \otimes \cdots \otimes W_j^{(n)}.$$

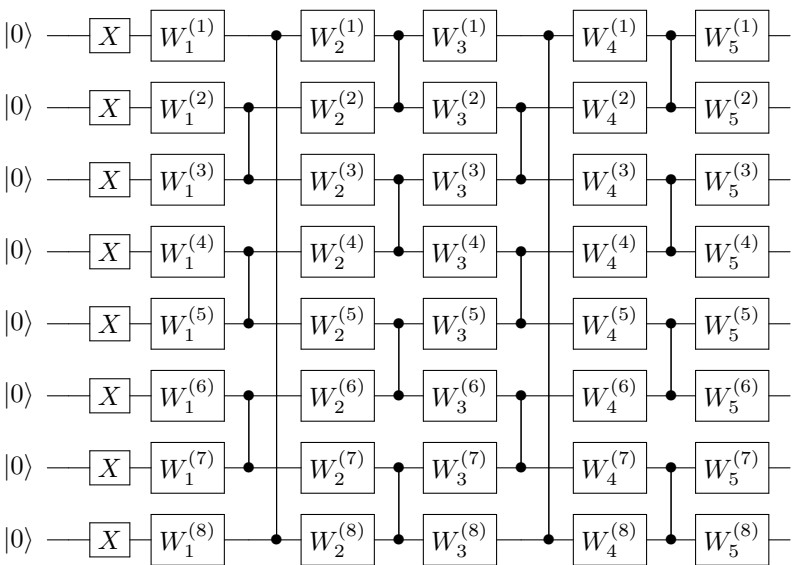

Figure 10: An example of the variational encoding model ($L = 2$ case).

Finally, we could mathematically define the encoding model:

$$U(\rho_{\text{in}}) = V_{2L+1} U_L U_{L-1} \cdots U_1 X^{\otimes n},$$

where $U_j = CZ_1 V_{2j} CZ_2 V_{2j-1}$ is the $j$-th alternating layer.

By employing the encoding model illustrated in Figure 10 for the state preparation, we find that the expectation of the term $\alpha(\rho_{\text{in}})$ defined in Theorem 1.1 has the lower bound independent from the qubit number.

**Theorem E.1.** *Suppose the input state $\rho_{in}(\boldsymbol{\theta})$ is prepared by the L-layer encoding model illustrated in Figure 10. Then,*

$$\mathbb{E}_{\boldsymbol{\beta}} \alpha(\rho_{in}) = \mathbb{E}_{\boldsymbol{\beta}} \left( Tr \left[ \sigma_{(1,0,\cdots,0)} \cdot \rho_{in} \right]^2 + Tr \left[ \sigma_{(3,0,\cdots,0)} \cdot \rho_{in} \right]^2 \right) \geq 2^{-2L},$$

*where $\boldsymbol{\beta}$ denotes all variational parameters in the encoding circuit, and the expectation is taken for all parameters in $\boldsymbol{\beta}$ with uniform distribution in $[0, 2\pi]$.*

*Proof.* Define $\rho_j = U_j U_{j-1} \cdots U_1 X^{\otimes n} |0\rangle^{\otimes n} \langle 0|^{\otimes n} X^{\otimes n} U_1^\dagger \cdots U_{j-1}^\dagger U_j^\dagger$, for $j \in \{0, 1, \cdots, L\}$. We have:

$$\mathbb{E}_{\boldsymbol{\beta}} \left( \text{Tr} \left[ \sigma_{(1,0,\cdots,0)} \cdot \rho_{\text{in}} \right]^2 + \text{Tr} \left[ \sigma_{(3,0,\cdots,0)} \cdot \rho_{\text{in}} \right]^2 \right) \tag{44}$$

$$= \mathbb{E}_{\boldsymbol{\beta}_1} \cdots \mathbb{E}_{\boldsymbol{\beta}_{2L+1}} \left( \text{Tr} \left[ \sigma_{(1,0,\cdots,0)} \cdot V_{2L+1} \rho_L V_{2L+1}^\dagger \right]^2 + \text{Tr} \left[ \sigma_{(3,0,\cdots,0)} \cdot V_{2L+1} \rho_L V_{2L+1}^\dagger \right]^2 \right) \tag{45}$$

$$= \mathbb{E}_{\boldsymbol{\beta}_1} \cdots \mathbb{E}_{\boldsymbol{\beta}_{2L}} \left( \text{Tr} \left[ \sigma_{(1,0,\cdots,0)} \cdot \rho_L \right]^2 + \text{Tr} \left[ \sigma_{(3,0,\cdots,0)} \cdot \rho_L \right]^2 \right). \tag{46}$$

$$\geq 2^{-2L} \left( \text{Tr} \left[ \sigma_{(1,0,\cdots,0)} \cdot \rho_0 \right]^2 + \text{Tr} \left[ \sigma_{(3,0,\cdots,0)} \cdot \rho_0 \right]^2 \right) \tag{47}$$

$$= 2^{-2L} \cdot (0^2 + (-1)^2) = 2^{-2L}, \tag{48}$$

where Eq. (45) is derived from the definition of $\rho_L$. Eq. (46) is derived using Lemma C.3. Eq. (47) is derived by noticing that for each $j \in \{0, 1, \cdots, L-1\}$, the following equations holds,

$$\mathbb{E}_{\boldsymbol{\beta}_1} \cdots \mathbb{E}_{\boldsymbol{\beta}_{2j+2}} \left( \text{Tr} \left[ \sigma_{(1,0,\cdots,0)} \cdot \rho_{j+1} \right]^2 + \text{Tr} \left[ \sigma_{(3,0,\cdots,0)} \cdot \rho_{j+1} \right]^2 \right) \tag{49}$$

$$= \mathbb{E}_{\boldsymbol{\beta}_1} \cdots \mathbb{E}_{\boldsymbol{\beta}_{2j+2}} \left( \text{Tr} \left[ \sigma_{(1,0,\cdots,0)} \cdot U_{j+1} \rho_j U_{j+1}^\dagger \right]^2 + \text{Tr} \left[ \sigma_{(3,0,\cdots,0)} \cdot U_{j+1} \rho_j U_{j+1}^\dagger \right]^2 \right) \tag{50}$$

$$= \mathbb{E}_{\boldsymbol{\beta}_1} \cdots \mathbb{E}_{\boldsymbol{\beta}_{2j+2}} \mathrm{Tr} \left[ \sigma_{(1,0,\cdots,0)} \cdot CZ_1 V_{2j+2} CZ_2 V_{2j+1} \rho_j V_{2j+1}^\dagger CZ_2 V_{2j+2}^\dagger CZ_1 \right]^2 \tag{51}$$

$$+ \mathbb{E}_{\boldsymbol{\beta}_1} \cdots \mathbb{E}_{\boldsymbol{\beta}_{2j+2}} \mathrm{Tr} \left[ \sigma_{(3,0,\cdots,0)} \cdot CZ_1 V_{2j+2} CZ_2 V_{2j+1} \rho_j V_{2j+1}^\dagger CZ_2 V_{2j+2}^\dagger CZ_1 \right]^2 \tag{52}$$

$$= \mathbb{E}_{\boldsymbol{\beta}_1} \cdots \mathbb{E}_{\boldsymbol{\beta}_{2j+2}} \mathrm{Tr} \left[ \sigma_{(1,3,0,\cdots,0)} \cdot V_{2j+2} CZ_2 V_{2j+1} \rho_j V_{2j+1}^\dagger CZ_2 V_{2j+2}^\dagger \right]^2 \tag{53}$$

$$+ \mathbb{E}_{\boldsymbol{\beta}_1} \cdots \mathbb{E}_{\boldsymbol{\beta}_{2j+2}} \mathrm{Tr} \left[ \sigma_{(3,0,\cdots,0)} \cdot V_{2j+2} CZ_2 V_{2j+1} \rho_j V_{2j+1}^\dagger CZ_2 V_{2j+2}^\dagger \right]^2 \tag{54}$$

$$\geq \mathbb{E}_{\boldsymbol{\beta}_1} \cdots \mathbb{E}_{\boldsymbol{\beta}_{2j+2}} \mathrm{Tr} \left[ \sigma_{(3,0,\cdots,0)} \cdot V_{2j+2} CZ_2 V_{2j+1} \rho_j V_{2j+1}^\dagger CZ_2 V_{2j+2}^\dagger \right]^2 \tag{55}$$

$$= \mathbb{E}_{\boldsymbol{\beta}_1} \cdots \mathbb{E}_{\boldsymbol{\beta}_{2j+1}} \left( \frac{1}{2} \mathrm{Tr} \left[ \sigma_{(1,0,\cdots,0)} \cdot CZ_2 V_{2j+1} \rho_j V_{2j+1}^\dagger CZ_2 \right]^2 + \frac{1}{2} \mathrm{Tr} \left[ \sigma_{(3,0,\cdots,0)} \cdot CZ_2 V_{2j+1} \rho_j V_{2j+1}^\dagger CZ_2 \right]^2 \right) \tag{56}$$

$$= \mathbb{E}_{\boldsymbol{\beta}_1} \cdots \mathbb{E}_{\boldsymbol{\beta}_{2j+1}} \left( \frac{1}{2} \mathrm{Tr} \left[ \sigma_{(1,0,\cdots,0,3)} \cdot V_{2j+1} \rho_j V_{2j+1}^\dagger \right]^2 + \frac{1}{2} \mathrm{Tr} \left[ \sigma_{(3,0,\cdots,0)} \cdot V_{2j+1} \rho_j V_{2j+1}^\dagger \right]^2 \right) \tag{57}$$

$$\geq \mathbb{E}_{\boldsymbol{\beta}_1} \cdots \mathbb{E}_{\boldsymbol{\beta}_{2j+1}} \frac{1}{2} \mathrm{Tr} \left[ \sigma_{(3,0,\cdots,0)} \cdot V_{2j+1} \rho_j V_{2j+1}^\dagger \right]^2 \tag{58}$$

$$= \mathbb{E}_{\boldsymbol{\beta}_1} \cdots \mathbb{E}_{\boldsymbol{\beta}_{2j}} \frac{1}{4} \left( \mathrm{Tr} \left[ \sigma_{(1,0,\cdots,0)} \cdot \rho_j \right]^2 + \mathrm{Tr} \left[ \sigma_{(3,0,\cdots,0)} \cdot \rho_j \right]^2 \right), \tag{59}$$

where Eq. (50) is derived from the definition of $\rho_{j+1}$. Eq. (51-52) are derived from the definition of $U_{j+1}$. Eq. (53-54) and Eq. (57) are derived using Lemma C.1. Eq. (56) and Eq. (59) are derived using Lemma C.3.

$\square$

## F  THE DEFORMED TREE TENSOR QNN

Similar to the TT-QNN case, the objective function for the Deformed Tree Tensor (DTT) QNN is given in the form:

$$f_{\mathrm{DTT}}(\boldsymbol{\theta}) = \frac{1}{2} + \frac{1}{2} \mathrm{Tr}[\sigma_3 \otimes I^{\otimes(n-1)} V_{\mathrm{DTT}}(\boldsymbol{\theta}) \rho_{\mathrm{in}} V_{\mathrm{DTT}}(\boldsymbol{\theta})^\dagger], \tag{60}$$

where $V_{\mathrm{DTT}}$ denotes the circuit operation of DTT-QNN which is illustrated in Figure 11. The lower bound result for the gradient norm of DTT-QNNs is provided in Theorem F.1.

**Theorem F.1.** *Consider the $n$-qubit DTT-QNN defined in Figure 11 and the corresponding objective function $f_{DTT}$ defined in (60), then we have:*

$$\frac{1 + \log n}{4n} \cdot \alpha(\rho_{in}) \leq \mathbb{E}_{\boldsymbol{\theta}} \|\nabla_{\boldsymbol{\theta}} f_{DTT}\|^2 \leq 2n - 1, \tag{61}$$

*where the expectation is taken for all parameters in $\boldsymbol{\theta}$ with uniform distributions in $[0, 2\pi]$, $\rho_{in} \in \mathbb{C}^{2^n \times 2^n}$ denotes the input state, $\alpha(\rho_{in}) = \mathrm{Tr} \left[ \sigma_{(1,0,\cdots,0)} \cdot \rho_{in} \right]^2 + \mathrm{Tr} \left[ \sigma_{(3,0,\cdots,0)} \cdot \rho_{in} \right]^2$, and $\sigma_{(i_1, i_2, \cdots, i_n)} \equiv \sigma_{i_1} \otimes \sigma_{i_2} \otimes \cdots \otimes \sigma_{i_n}$.*

*Proof.* Firstly we remark that by Lemma D.1, each partial derivative is calculated as

$$\frac{\partial f_{\mathrm{DTT}}}{\partial \theta_j^{(k)}} = \frac{1}{2} \left( \mathrm{Tr}[O \cdot V_{\mathrm{DTT}}(\boldsymbol{\theta}_+) \rho_{\mathrm{in}} V_{\mathrm{DTT}}(\boldsymbol{\theta}_+)^\dagger] - \mathrm{Tr}[O \cdot V_{\mathrm{DTT}}(\boldsymbol{\theta}_-) \rho_{\mathrm{in}} V_{\mathrm{DTT}}(\boldsymbol{\theta}_-)^\dagger] \right),$$

since the expectation of the quantum observable is bounded by $[-1, 1]$, the square of the partial derivative could be easily bounded as:

$$\left( \frac{\partial f_{\mathrm{DTT}}}{\partial \theta_j^{(k)}} \right)^2 \leq 1.$$

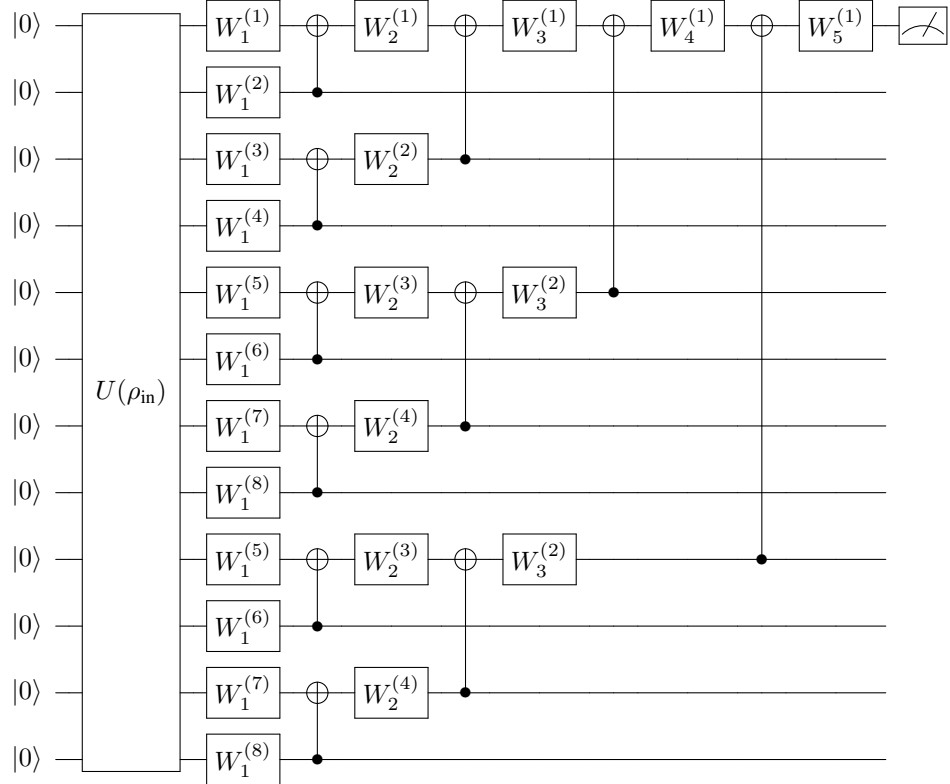

Figure 11: Quantum Neural Network with the Deformed Tree Tensor structure (qubit number = 12).

By summing up $2n - 1$ parameters, we obtain

$$\|\nabla_{\boldsymbol{\theta}} f_{\text{DTT}}\|^2 = \sum_{j,k} \left( \frac{\partial f_{\text{DTT}}}{\partial \theta_j^{(k)}} \right)^2 \leq 2n - 1.$$

On the other side, the lower bound could be derived as follows,

$$\mathbb{E}_{\boldsymbol{\theta}} \|\nabla_{\boldsymbol{\theta}} f_{\text{DTT}}\|^2 \geq \sum_{j=1}^{1+\lceil \log n \rceil} \mathbb{E}_{\boldsymbol{\theta}} \left( \frac{\partial f_{\text{DTT}}}{\partial \theta_j^{(1)}} \right)^2 \tag{62}$$

$$= \sum_{j=1}^{1+\lceil \log n \rceil} 4\mathbb{E}_{\boldsymbol{\theta}} \left( f_{\text{DTT}} - \frac{1}{2} \right)^2 \tag{63}$$

$$\geq \frac{1 + \log n}{4n} \cdot \left( \text{Tr} \left[ \sigma_{(1,0,\cdots,0)} \cdot \rho_{\text{in}} \right]^2 + \text{Tr} \left[ \sigma_{(3,0,\cdots,0)} \cdot \rho_{\text{in}} \right]^2 \right), \tag{64}$$

where Eq. (63) is derived using Lemma F.1, and Eq. (64) is derived using Lemma F.2.

$\square$

**Lemma F.1.** *For the objective function $f_{DTT}$ defined in Eq. (60), the following formula holds for every $j \in \{1, 2, \cdots, 1 + \lceil \log n \rceil\}$:*

$$\mathbb{E}_{\boldsymbol{\theta}} \left( \frac{\partial f_{DTT}}{\partial \theta_j^{(1)}} \right)^2 = 4 \cdot \mathbb{E}_{\boldsymbol{\theta}} (f_{DTT} - \frac{1}{2})^2, \tag{65}$$

*where the expectation is taken for all parameters in $\theta$ with uniform distribution in $[0, 2\pi]$.*

*Proof.* The proof has a very similar formulation compare to the original tree tensor case. First we rewrite the formulation of $f_{\text{DTT}}$ in detail:

$$f_{\text{DTT}} = \frac{1}{2} + \frac{1}{2}\text{Tr}\left[\sigma_{(3,0,\cdots,0)} \cdot V_{m+1}CX_m \cdots CX_1V_1 \cdot \rho_{\text{in}} \cdot V_1^\dagger CX_1^\dagger \cdots CX_m^\dagger V_{m+1}^\dagger\right], \quad (66)$$

where $m = \lceil \log n \rceil$ and we denote

$$\sigma_{(i_1,i_2,\cdots,i_n)} \equiv \sigma_{i_1} \otimes \sigma_{i_2} \otimes \cdots \otimes \sigma_{i_n}.$$

The operation $V_\ell$ consists of $\lfloor n \cdot 2^{1-\ell} \rfloor$ single qubit rotations $W_\ell^{(j)} = e^{-i\sigma_2\theta_\ell^{(j)}}$ on the $(j-1) \cdot 2^{\ell-1} + 1$-th qubit, where $j = 1, 2, \cdots, \lfloor n \cdot 2^{1-\ell} \rfloor$. The operation $CX_\ell$ consists of $\lfloor n \cdot 2^{-\ell} \rfloor$ CNOT gates, where each of them acts on the $(j-1) \cdot 2^\ell + 1$-th and $(j - 0.5) \cdot 2^\ell + 1$-th qubit, for $j = 1, 2, \cdots, \lfloor n \cdot 2^{-\ell} \rfloor$.

Now we focus on the partial derivative of the function $f$ to the parameter $\theta_j^{(1)}$. We have:

$$\frac{\partial f_{\text{DTT}}}{\partial \theta_j^{(1)}} = \frac{1}{2}\text{Tr}\left[\sigma_{(3,0,\cdots,0)} \cdot V_{m+1}CX_m \cdots \frac{\partial V_j}{\theta_j^{(1)}} \cdots CX_1V_1 \cdot \rho_{\text{in}} \cdot V_1^\dagger CX_1^\dagger \cdots V_j^\dagger \cdots CX_m^\dagger V_{m+1}^\dagger\right] \quad (67)$$

$$+ \frac{1}{2}\text{Tr}\left[\sigma_{(3,0,\cdots,0)} \cdot V_{m+1}CX_m \cdots V_j \cdots CX_1V_1 \cdot \rho_{\text{in}} \cdot V_1^\dagger CX_1^\dagger \cdots \frac{\partial V_j^\dagger}{\theta_j^{(1)}} \cdots CX_m^\dagger V_{m+1}^\dagger\right] \quad (68)$$

$$= \text{Tr}\left[\sigma_{(3,0,\cdots,0)} \cdot V_{m+1}CX_m \cdots \frac{\partial V_j}{\theta_j^{(1)}} \cdots CX_1V_1 \cdot \rho_{\text{in}} \cdot V_1^\dagger CX_1^\dagger \cdots V_j^\dagger \cdots CX_m^\dagger V_{m+1}^\dagger\right]. \quad (69)$$

The Eq. (69) holds because both terms in (67) and (68) except $\rho_{\text{in}}$ are real matrices, and $\rho_{\text{in}} = \rho_{\text{in}}^\dagger$. Similar to the tree tensor case, the key idea to derive $\mathbb{E}_{\boldsymbol{\theta}}\left(\frac{\partial f_{\text{DTT}}}{\partial \theta_j^{(1)}}\right)^2 = 4 \cdot \mathbb{E}_\theta(f_{\text{DTT}} - \frac{1}{2})^2$ is that for cases $B = D = \sigma_j \in \{\sigma_1, \sigma_3\}$, the term $\frac{\delta_{j0}+\delta_{j2}}{2} = 0$, which means Lemma C.3 and Lemma C.4 collapse to the same formulation:

$$\mathbb{E}_\theta\text{Tr}[WAW^\dagger B]\text{Tr}[WCW^\dagger D] = \mathbb{E}_\theta\text{Tr}[GAW^\dagger B]\text{Tr}[GCW^\dagger D]$$
$$= \frac{1}{2}\text{Tr}[A\sigma_1]\text{Tr}[C\sigma_1] + \frac{1}{2}\text{Tr}[A\sigma_3]\text{Tr}[C\sigma_3].$$

Now we write the analysis in detail.

$$\mathbb{E}_{\boldsymbol{\theta}}\left[\left(\frac{\partial f_{\text{DTT}}}{\partial \theta_j^{(1)}}\right)^2 - 4(f_{\text{DTT}} - \frac{1}{2})^2\right] \quad (70)$$

$$=\mathbb{E}_{\boldsymbol{\theta}_1} \cdots \mathbb{E}_{\boldsymbol{\theta}_m}\mathbb{E}_{\theta_{m+1}^{(1)}} \text{Tr}\left[\sigma_{(3,0,\cdots,0)} \cdot V_{m+1}CX_mA_mCX_mV_{m+1}^\dagger\right]^2 \quad (71)$$

$$-\mathbb{E}_{\boldsymbol{\theta}_1} \cdots \mathbb{E}_{\boldsymbol{\theta}_m}\mathbb{E}_{\theta_{m+1}^{(1)}} \text{Tr}\left[\sigma_{(3,0,\cdots,0)} \cdot V_{m+1}CX_mB_mCX_mV_{m+1}^\dagger\right]^2 \quad (72)$$

$$=\mathbb{E}_{\boldsymbol{\theta}_1} \cdots \mathbb{E}_{\boldsymbol{\theta}_m}\left\{\frac{1}{2}\text{Tr}\left[\sigma_{(3,0,\cdots,0,3,0,\cdots,0)} \cdot A_m\right]^2 + \frac{1}{2}\text{Tr}\left[\sigma_{(1,0,\cdots,0,0,0,\cdots,0)} \cdot A_m\right]^2\right\} \quad (73)$$

$$-\mathbb{E}_{\boldsymbol{\theta}_1} \cdots \mathbb{E}_{\boldsymbol{\theta}_m}\left\{\frac{1}{2}\text{Tr}\left[\sigma_{(3,0,\cdots,0,3,0,\cdots,0)} \cdot B_m\right]^2 + \frac{1}{2}\text{Tr}\left[\sigma_{(1,0,\cdots,0,0,0,\cdots,0)} \cdot B_m\right]^2\right\}, \quad (74)$$

where

$$A_m = V_mCX_{m-1} \cdots \frac{\partial V_j}{\theta_j^{(1)}} \cdots CX_1V_1 \cdot \rho_{\text{in}} \cdot V_1^\dagger CX_1^\dagger \cdots V_j^\dagger \cdots CX_{m-1}^\dagger V_m^\dagger,$$

$$B_m = V_m CX_{m-1} \cdots V_j \cdots CX_1 V_1 \cdot \rho_{\text{in}} \cdot V_1^\dagger CX_1^\dagger \cdots V_j^\dagger \cdots CX_{m-1}^\dagger V_m^\dagger.$$

Eq. (73) and Eq. (74) are derived using the collapsed form of Lemma C.1:

$$\text{CNOT}(\sigma_1 \otimes \sigma_0)\text{CNOT}^\dagger = \sigma_1 \otimes \sigma_0, \ \text{CNOT}(\sigma_3 \otimes \sigma_0)\text{CNOT}^\dagger = \sigma_3 \otimes \sigma_3,$$

and $\boldsymbol{\theta}_\ell$ denotes the vector consisted with all parameters in the $\ell$-th layer. The integration could be performed for parameters $\{\boldsymbol{\theta}_m, \boldsymbol{\theta}_{m-1}, \cdots, \boldsymbol{\theta}_{j+1}\}$. It is not hard to find that after the integration of the parameters $\boldsymbol{\theta}_{j+1}$, the term $\text{Tr}[\sigma_{\boldsymbol{i}} \cdot A_j]^2$ and $\text{Tr}[\sigma_{\boldsymbol{i}} \cdot B_j]^2$ have the opposite coefficients. Besides, the first index of each Pauli tensor product $\sigma_{(i_1, i_2, \cdots, i_n)}$ could only be $i_1 \in \{1, 3\}$ because of the Lemma C.3. So we could write

$$\mathbb{E}_{\boldsymbol{\theta}} \left[ \left( \frac{\partial f_{\text{DTT}}}{\partial \theta_j^{(1)}} \right)^2 - 4(f_{\text{DTT}} - \frac{1}{2})^2 \right] \tag{75}$$

$$= \mathbb{E}_{\boldsymbol{\theta}_1} \cdots \mathbb{E}_{\boldsymbol{\theta}_j} \left\{ \sum_{i_1 \in \{1,3\}} \sum_{i_2=0}^{3} \cdots \sum_{i_n=0}^{3} a_{\boldsymbol{i}} \text{Tr}\left[\sigma_{\boldsymbol{i}} \cdot A_j\right]^2 - a_{\boldsymbol{i}} \text{Tr}\left[\sigma_{\boldsymbol{i}} \cdot B_j\right]^2 \right\} \tag{76}$$

where

$$A_j = \frac{\partial V_j}{\partial \theta_j^{(1)}} CX_{j-1} \cdots CX_1 V_1 \cdot \rho_{\text{in}} \cdot V_1^\dagger CX_1^\dagger \cdots CX_{j-1} V_j^\dagger$$

$$= (G_j^{(1)} \otimes I^{\otimes(n-1)}) A_j^{/\theta_j^{(1)}} (W_j^{(1)\dagger} \otimes I^{\otimes(n-1)}),$$

$$B_j = V_j CX_{j-1} \cdots CX_1 V_1 \cdot \rho_{\text{in}} \cdot V_1^\dagger CX_1^\dagger \cdots CX_{j-1} V_j^\dagger$$

$$= (W_j^{(1)} \otimes I^{\otimes(n-1)}) A_j^{/\theta_j^{(1)}} (W_j^{(1)\dagger} \otimes I^{\otimes(n-1)}),$$

and $a_{\boldsymbol{i}}$ is the coefficient of the term $\text{Tr}\left[\sigma_{\boldsymbol{i}} \cdot A_j\right]^2$. We denote $G_j^{(1)} = \frac{\partial W_j^{(1)}}{\partial \theta_j^{(1)}}$ and use $A_j^{/\theta_j^{(1)}}$ to denote the rest part of $A_j$ and $B_j$. By Lemma C.3 and Lemma C.4, we have

$$\mathbb{E}_{\theta_j^{(1)}} \left[ \text{Tr}\left[\sigma_{\boldsymbol{i}} \cdot A_j\right]^2 - \text{Tr}\left[\sigma_{\boldsymbol{i}} \cdot B_j\right]^2 \right] = 0,$$

since for the case $i_1 \in \{1, 3\}$, the term $\frac{\delta_{i_1 0} + \delta_{i_1 2}}{2} = 0$, which means Lemma C.3 and Lemma C.4 have the same formulation. Then, we derive the Eq. (65).

$\square$

**Lemma F.2.** *For the loss function $f_{DTT}$ defined in (60), we have:*

$$\mathbb{E}_{\boldsymbol{\theta}}(f_{DTT} - \frac{1}{2})^2 \geq \frac{\left\{Tr\left[\sigma_{(1,0,\cdots,0)} \cdot \rho_{in}\right]\right\}^2 + \left\{Tr\left[\sigma_{(3,0,\cdots,0)} \cdot \rho_{in}\right]\right\}^2}{16n}, \tag{77}$$

*where we denote*

$$\sigma_{(i_1, i_2, \cdots, i_n)} \equiv \sigma_{i_1} \otimes \sigma_{i_2} \otimes \cdots \otimes \sigma_{i_n},$$

*and the expectation is taken for all parameters in $\theta$ with uniform distributions in $[0, 2\pi]$.*

*Proof.* First we expand the function $f_{\text{DTT}}$ in detail,

$$f_{\text{TT}} = \frac{1}{2} + \frac{1}{2}\text{Tr}\left[\sigma_{(3,0,\cdots,0)} \cdot V_{m+1} CX_m \cdots CX_1 V_1 \cdot \rho_{\text{in}} \cdot V_1^\dagger CX_1^\dagger \cdots CX_m^\dagger V_{m+1}^\dagger\right], \tag{78}$$

where $m = \lceil \log n \rceil$. Now we consider the expectation of $(f_{\text{DTT}} - \frac{1}{2})^2$ under the uniform distribution for $\theta_{m+1}^{(1)}$:

$$\mathbb{E}_{\theta_{m+1}^{(1)}}(f_{\text{DTT}} - \frac{1}{2})^2 = \frac{1}{4} \mathbb{E}_{\theta_{m+1}^{(1)}} \left\{ \text{Tr}\left[\sigma_{(3,0,\cdots,0)} \cdot V_{m+1} CX_m \cdots CX_1 V_1 \cdot \rho_{\text{in}} \cdot V_1^\dagger CX_1^\dagger \cdots CX_m^\dagger V_{m+1}^\dagger\right] \right\}^2 \tag{79}$$

$$= \frac{1}{8} \left\{ \text{Tr}\left[\sigma_{(3,0,\cdots,0)} \cdot A'\right] \right\}^2 + \frac{1}{8} \left\{ \text{Tr}\left[\sigma_{(1,0,\cdots,0)} \cdot A'\right] \right\}^2 \tag{80}$$

$$= \frac{1}{8} \left\{ \mathrm{Tr} \left[ \sigma_{(3,0,\cdots,0,3,0,\cdots,0)} \cdot A \right] \right\}^2 + \frac{1}{8} \left\{ \mathrm{Tr} \left[ \sigma_{(1,0,\cdots,0,0,0,\cdots,0)} \cdot A \right] \right\}^2 \qquad (81)$$

$$\geq \frac{1}{8} \left\{ \mathrm{Tr} \left[ \sigma_{(1,0,\cdots,0,0,0,\cdots,0)} \cdot A \right] \right\}^2, \qquad (82)$$

where

$$A' = CX_m V_m \cdots CX_1 V_1 \cdot \rho_{\mathrm{in}} \cdot V_1^\dagger CX_1^\dagger \cdots V_m CX_m^\dagger, \qquad (83)$$

$$A = V_m \cdots CX_1 V_1 \cdot \rho_{\mathrm{in}} \cdot V_1^\dagger CX_1^\dagger \cdots V_m. \qquad (84)$$

Eq. (80) is derived using Lemma C.3, and Eq. (81) is derived using the collapsed form of Lemma C.1:

$$\mathrm{CNOT}(\sigma_1 \otimes \sigma_0)\mathrm{CNOT}^\dagger = \sigma_1 \otimes \sigma_0, \ \mathrm{CNOT}(\sigma_3 \otimes \sigma_0)\mathrm{CNOT}^\dagger = \sigma_3 \otimes \sigma_3,$$

We remark that during the integration of the parameters $\{\theta_\ell^{(j)}\}$ in each layer $\ell \in \{1, 2, \cdots, m+1\}$, the coefficient of the term $\{\mathrm{Tr}[\sigma_{(1,0,\cdots,0)} \cdot A]\}^2$ only times a factor $1/2$ for the case $j = 1$, and the coefficient remains for the cases $j > 1$ (check Lemma C.3 for detail). Since the formulation $\{\mathrm{Tr}[\sigma_{(1,0,\cdots,0)} \cdot A]\}^2$ remains the same when merging the operation $CX_\ell$ with $\sigma_{(1,0,\cdots,0)}$, for $\ell \in \{1, 2, \cdots, m\}$, we could generate the following equation,

$$\mathbb{E}_{\boldsymbol{\theta}_2} \cdots \mathbb{E}_{\boldsymbol{\theta}_{m+1}} (f_{\mathrm{DTT}} - \frac{1}{2})^2 \geq (\frac{1}{2})^{m+2} \left\{ \mathrm{Tr} \left[ \sigma_{(1,0,\cdots,0)} \cdot V_1 \cdot \rho_{\mathrm{in}} \cdot V_1^\dagger \right] \right\}^2, \qquad (85)$$

where $\boldsymbol{\theta}_\ell$ denotes the vector consisted with all parameters in the $\ell$-th layer.

Finally by using Lemma C.3, we could integrate the parameters $\{\theta_1^{(j)}\}_{j=1}^n$ in (85):

$$\mathbb{E}_{\boldsymbol{\theta}}(f_{\mathrm{DTT}} - \frac{1}{2})^2 = \mathbb{E}_{\boldsymbol{\theta}_1} \cdots \mathbb{E}_{\boldsymbol{\theta}_{m+1}}(f_{\mathrm{DTT}} - \frac{1}{2})^2 \qquad (86)$$

$$\geq \frac{\left\{ \mathrm{Tr} \left[ \sigma_{(1,0,\cdots,0)} \cdot \rho_{\mathrm{in}} \right] \right\}^2 + \left\{ \mathrm{Tr} \left[ \sigma_{(3,0,\cdots,0)} \cdot \rho_{\mathrm{in}} \right] \right\}^2}{2^{m+3}} \qquad (87)$$

$$\geq \frac{\left\{ \mathrm{Tr} \left[ \sigma_{(1,0,\cdots,0)} \cdot \rho_{\mathrm{in}} \right] \right\}^2 + \left\{ \mathrm{Tr} \left[ \sigma_{(3,0,\cdots,0)} \cdot \rho_{\mathrm{in}} \right] \right\}^2}{16n}. \qquad (88)$$

$$\square$$

## G   THE PROOF OF THEOREM 3.1: THE SC PART

**Theorem G.1.** *Consider the $n$-qubit SC-QNN defined in Figure 2 and the corresponding objective function $f_{SC}$ defined in (4), then we have:*

$$\frac{1 + n_c}{2^{1+n_c}} \cdot \alpha(\rho_{in}) \leq \mathbb{E}_{\boldsymbol{\theta}} \|\nabla_{\boldsymbol{\theta}} f_{SC}\|^2 \leq 2n - 1, \qquad (89)$$

*where $n_c$ is the number of the control operation CNOT that directly links to the first qubit channel, and the expectation is taken for all parameters in $\boldsymbol{\theta}$ with uniform distributions in $[0, 2\pi]$, $\rho_{in} \in \mathbb{C}^{2^n \times 2^n}$ denotes the input state, $\alpha(\rho_{in}) = Tr \left[ \sigma_{(1,0,\cdots,0)} \cdot \rho_{in} \right]^2 + Tr \left[ \sigma_{(3,0,\cdots,0)} \cdot \rho_{in} \right]^2$, and $\sigma_{(i_1,i_2,\cdots,i_n)} \equiv \sigma_{i_1} \otimes \sigma_{i_2} \otimes \cdots \otimes \sigma_{i_n}$.*

*Proof.* Firstly we remark that by Lemma D.1, each partial derivative is calculated as

$$\frac{\partial f_{\mathrm{SC}}}{\partial \theta_j^{(k)}} = \frac{1}{2} \left( \mathrm{Tr}[O \cdot V_{\mathrm{SC}}(\boldsymbol{\theta}_+)\rho_{\mathrm{in}}V_{\mathrm{SC}}(\boldsymbol{\theta}_+)^\dagger] - \mathrm{Tr}[O \cdot V_{\mathrm{SC}}(\boldsymbol{\theta}_-)\rho_{\mathrm{in}}V_{\mathrm{SC}}(\boldsymbol{\theta}_-)^\dagger] \right),$$

since the expectation of the quantum observable is bounded by $[-1, 1]$, the square of the partial derivative could be easily bounded as:

$$\left( \frac{\partial f_{\mathrm{SC}}}{\partial \theta_j^{(k)}} \right)^2 \leq 1.$$

By summing up $2n - 1$ parameters, we obtain

$$\|\nabla_{\boldsymbol{\theta}} f_{\text{SC}}\|^2 = \sum_{j,k} \left( \frac{\partial f_{\text{SC}}}{\partial \theta_j^{(k)}} \right)^2 \leq 2n - 1.$$

On the other side, the lower bound could be derived as follows,

$$\mathbb{E}_{\boldsymbol{\theta}} \|\nabla_{\boldsymbol{\theta}} f_{\text{SC}}\|^2 \geq \mathbb{E}_{\boldsymbol{\theta}} \left( \frac{\partial f_{\text{SC}}}{\partial \theta_1^{(1)}} \right)^2 + \sum_{j=n-n_c+1}^{n} \mathbb{E}_{\boldsymbol{\theta}} \left( \frac{\partial f_{\text{SC}}}{\partial \theta_j^{(1)}} \right)^2 \tag{90}$$

$$= (1 + n_c) \cdot 4 \cdot \mathbb{E}_{\boldsymbol{\theta}} \left( f_{\text{SC}} - \frac{1}{2} \right)^2 \tag{91}$$

$$\geq \frac{1 + n_c}{2^{1+n_c}} \cdot \left( \text{Tr} \left[ \sigma_{(1,0,\cdots,0)} \cdot \rho_{\text{in}} \right]^2 + \text{Tr} \left[ \sigma_{(3,0,\cdots,0)} \cdot \rho_{\text{in}} \right]^2 \right), \tag{92}$$

where Eq. (91) is derived using Lemma G.1, and Eq. (92) is derived using Lemma G.2.

$\square$

**Lemma G.1.** *For the objective function $f_{SC}$ defined in Eq. (60), the following formula holds for every $j$ such that $\theta_j^{(1)}$ tunes the single qubit gate on the first qubit channel:*

$$\mathbb{E}_{\boldsymbol{\theta}} \left( \frac{\partial f_{SC}}{\partial \theta_j^{(1)}} \right)^2 = 4 \cdot \mathbb{E}_{\boldsymbol{\theta}} \left( f_{SC} - \frac{1}{2} \right)^2, \tag{93}$$

*where the expectation is taken for all parameters in $\boldsymbol{\theta}$ with uniform distribution in $[0, 2\pi]$.*

*Proof.* First we write the formulation of $f_{SC}$ in detail:

$$f_{\text{SC}} = \frac{1}{2} + \frac{1}{2} \text{Tr} \left[ \sigma_{(3,0,\cdots,0)} \cdot V_n CX_{n-1} \cdots CX_1 V_1 \cdot \rho_{\text{in}} \cdot V_1^\dagger CX_1^\dagger \cdots CX_{n-1}^\dagger V_n^\dagger \right], \tag{94}$$

where we denote $\sigma_{(i_1, i_2, \cdots, i_n)} \equiv \sigma_{i_1} \otimes \sigma_{i_2} \otimes \cdots \otimes \sigma_{i_n}$. The operation $CX_\ell$ is defined as,

$$CX_\ell = \begin{cases} \text{CNOT operation on qubits pair } (n+1-\ell, n-\ell) & (1 \leq \ell \leq n-1-n_c), \\ \text{CNOT operation on qubits pair } (n+1-\ell, 1) & (n - n_c \leq \ell \leq n-1). \end{cases}$$

The operation $V_\ell$ is defined as,

$$V_\ell = \begin{cases} W_1^{(1)} \otimes W_1^{(2)} \otimes \cdots \otimes W_1^{(n)} & (\ell = 1), \\ I^{\otimes (n-\ell)} \otimes W_\ell^{(1)} \otimes I^{\otimes (\ell-1)} & (1 \leq \ell \leq n-1-n_c), \\ W_\ell^{(1)} \otimes I \otimes I \otimes \cdots \otimes I & (n - n_c \leq \ell \leq n). \end{cases}$$

Now we focus on the partial derivative of the function $f_{\text{SC}}$ to the parameter $\theta_j^{(1)}$. We have:

$$\frac{\partial f_{\text{SC}}}{\partial \theta_j^{(1)}} = \frac{1}{2} \text{Tr} \left[ \sigma_{(3,0,\cdots,0)} \cdot V_n CX_{n-1} \cdots \frac{\partial V_j}{\theta_j^{(1)}} \cdots CX_1 V_1 \cdot \rho_{\text{in}} \cdot V_1^\dagger CX_1^\dagger \cdots V_j^\dagger \cdots CX_{n-1}^\dagger V_n^\dagger \right] \tag{95}$$

$$+ \frac{1}{2} \text{Tr} \left[ \sigma_{(3,0,\cdots,0)} \cdot V_n CX_{n-1} \cdots V_j \cdots CX_1 V_1 \cdot \rho_{\text{in}} \cdot V_1^\dagger CX_1^\dagger \cdots \frac{\partial V_j^\dagger}{\theta_j^{(1)}} \cdots CX_{n-1}^\dagger V_n^\dagger \right] \tag{96}$$

$$= \text{Tr} \left[ \sigma_{(3,0,\cdots,0)} \cdot V_n CX_{n-1} \cdots \frac{\partial V_j}{\theta_j^{(1)}} \cdots CX_1 V_1 \cdot \rho_{\text{in}} \cdot V_1^\dagger CX_1^\dagger \cdots V_j^\dagger \cdots CX_{n-1}^\dagger V_n^\dagger \right]. \tag{97}$$

The Eq. (97) holds because both terms in (95) and (96) except $\rho_{\text{in}}$ are real matrices, and $\rho_{\text{in}} = \rho_{\text{in}}^\dagger$. The key idea to derive $\mathbb{E}_{\boldsymbol{\theta}} \left( \frac{\partial f_{\text{SC}}}{\partial \theta_j^{(1)}} \right)^2 = 4 \cdot \mathbb{E}_\theta (f_{\text{SC}} - \frac{1}{2})^2$ is that for cases $B = D = \sigma_j \in \{\sigma_1, \sigma_3\}$ in Lemma C.3 and Lemma C.4, the term $\frac{\delta_{j0} + \delta_{j2}}{2} = 0$, which means both lemma collapse to the same formulation:

$$
\begin{aligned}
\mathbb{E}_\theta \text{Tr}[W A W^\dagger B] \text{Tr}[W C W^\dagger D] &= \mathbb{E}_\theta \text{Tr}[G A W^\dagger B] \text{Tr}[G C W^\dagger D] \\
&= \frac{1}{2} \text{Tr}[A\sigma_1] \text{Tr}[C\sigma_1] + \frac{1}{2} \text{Tr}[A\sigma_3] \text{Tr}[C\sigma_3].
\end{aligned}
$$

Now we write the analysis in detail.

$$
\mathbb{E}_{\boldsymbol{\theta}} \left[ \left( \frac{\partial f_{\text{SC}}}{\partial \theta_j^{(1)}} \right)^2 - 4 \left( f_{\text{SC}} - \frac{1}{2} \right)^2 \right] \tag{98}
$$

$$
= \mathbb{E}_{\boldsymbol{\theta}_1} \cdots \mathbb{E}_{\boldsymbol{\theta}_{n-1}} \mathbb{E}_{\boldsymbol{\theta}_n} \text{Tr} \left[ \sigma_{(3,0,\cdots,0)} \cdot V_n C X_{n-1} A_{n-1} C X_{n-1} V_n^\dagger \right]^2 \tag{99}
$$

$$
- \mathbb{E}_{\boldsymbol{\theta}_1} \cdots \mathbb{E}_{\boldsymbol{\theta}_{n-1}} \mathbb{E}_{\boldsymbol{\theta}_n} \text{Tr} \left[ \sigma_{(3,0,\cdots,0)} \cdot V_n C X_{n-1} B_{n-1} C X_{n-1} V_n^\dagger \right]^2 \tag{100}
$$

$$
= \mathbb{E}_{\boldsymbol{\theta}_1} \cdots \mathbb{E}_{\boldsymbol{\theta}_{n-1}} \left\{ \frac{1}{2} \text{Tr} \left[ \sigma_{(3,3,0,\cdots,0)} \cdot A_{n-1} \right]^2 + \frac{1}{2} \text{Tr} \left[ \sigma_{(1,0,0,\cdots,0)} \cdot A_{n-1} \right]^2 \right\} \tag{101}
$$

$$
- \mathbb{E}_{\boldsymbol{\theta}_1} \cdots \mathbb{E}_{\boldsymbol{\theta}_{n-1}} \left\{ \frac{1}{2} \text{Tr} \left[ \sigma_{(3,3,0,\cdots,0)} \cdot B_{n-1} \right]^2 + \frac{1}{2} \text{Tr} \left[ \sigma_{(1,0,0,\cdots,0)} \cdot B_{n-1} \right]^2 \right\}, \tag{102}
$$

where

$$
A_{n-1} = V_{n-1} C X_{n-2} \cdots \frac{\partial V_j}{\theta_j^{(1)}} \cdots C X_1 V_1 \cdot \rho_{\text{in}} \cdot V_1^\dagger C X_1^\dagger \cdots V_j^\dagger \cdots C X_{n-2}^\dagger V_{n-1}^\dagger,
$$

$$
B_{n-1} = V_{n-1} C X_{n-2} \cdots V_j \cdots C X_1 V_1 \cdot \rho_{\text{in}} \cdot V_1^\dagger C X_1^\dagger \cdots V_j^\dagger \cdots C X_{n-2}^\dagger V_{n-1}^\dagger.
$$

Eq. (101) and Eq. (102) are derived using Lemma C.3 and the collapsed form of Lemma C.1:

$$
\text{CNOT}(\sigma_1 \otimes \sigma_0)\text{CNOT}^\dagger = \sigma_1 \otimes \sigma_0, \ \text{CNOT}(\sigma_3 \otimes \sigma_0)\text{CNOT}^\dagger = \sigma_3 \otimes \sigma_3,
$$

and $\boldsymbol{\theta}_\ell$ denotes the vector consisted with all parameters in the $\ell$-th layer. The integration (99)-(102) could be performed similarly for parameters $\{\boldsymbol{\theta}_{n-1}, \boldsymbol{\theta}_{n-2}, \cdots, \boldsymbol{\theta}_{j+1}\}$. It is not hard to find that after the integration of the parameters $\boldsymbol{\theta}_{j+1}$, the term $\text{Tr}[\sigma_i \cdot A_j]^2$ and $\text{Tr}[\sigma_i \cdot B_j]^2$ have the opposite coefficients. Besides, the first index of each Pauli tensor product $\sigma_{(i_1, i_2, \cdots, i_n)}$ could only be $i_1 \in \{1, 3\}$ because of the Lemma C.3. So we could write

$$
\mathbb{E}_{\boldsymbol{\theta}} \left[ \left( \frac{\partial f_{\text{SC}}}{\partial \theta_j^{(1)}} \right)^2 - 4 \left( f_{\text{SC}} - \frac{1}{2} \right)^2 \right] \tag{103}
$$

$$
= \mathbb{E}_{\boldsymbol{\theta}_1} \cdots \mathbb{E}_{\boldsymbol{\theta}_j} \left\{ \sum_{i_1 \in \{1,3\}} \sum_{i_2=0}^{3} \cdots \sum_{i_n=0}^{3} a_i \text{Tr} \left[ \sigma_i \cdot A_j \right]^2 - a_i \text{Tr} \left[ \sigma_i \cdot B_j \right]^2 \right\}, \tag{104}
$$

where

$$
\begin{aligned}
A_j &= \frac{\partial V_j}{\partial \theta_j^{(1)}} C X_{j-1} \cdots C X_1 V_1 \cdot \rho_{\text{in}} \cdot V_1^\dagger C X_1^\dagger \cdots C X_{j-1} V_j^\dagger \\
&= (G_j^{(1)} \otimes I^{\otimes (n-1)}) A_j^{\prime \theta_j^{(1)}} (W_j^{(1)\dagger} \otimes I^{\otimes (n-1)}), \\
B_j &= V_j C X_{j-1} \cdots C X_1 V_1 \cdot \rho_{\text{in}} \cdot V_1^\dagger C X_1^\dagger \cdots C X_{j-1} V_j^\dagger \\
&= (W_j^{(1)} \otimes I^{\otimes (n-1)}) A_j^{\prime \theta_j^{(1)}} (W_j^{(1)\dagger} \otimes I^{\otimes (n-1)}),
\end{aligned}
$$

and $\pm a_{\boldsymbol{i}}$ are coefficients of the term $\text{Tr}\,[\sigma_{\boldsymbol{i}} \cdot A_j]^2$, $\text{Tr}\,[\sigma_{\boldsymbol{i}} \cdot B_j]^2$, respectively. We denote $G_j^{(1)} = \frac{\partial W_j^{(1)}}{\partial \theta_j^{(1)}}$ and use $A_j^{'\theta_j^{(1)}}$ to denote the rest part of $A_j$ and $B_j$. By Lemma C.3 and Lemma C.4, we have

$$\mathbb{E}_{\theta_j^{(1)}} \left[ \text{Tr}\,[\sigma_{\boldsymbol{i}} \cdot A_j]^2 - \text{Tr}\,[\sigma_{\boldsymbol{i}} \cdot B_j]^2 \right] = 0,$$

since for the case $i_1 \in \{1,3\}$, the term $\frac{\delta_{i_1 0} + \delta_{i_1 2}}{2} = 0$ in Lemma C.3 and Lemma C.4, which means both lemmas have the same formulation. Then, we derive the Eq. (93).

$\square$

**Lemma G.2.** *For the loss function $f_{SC}$ defined in (4), we have:*

$$\mathbb{E}_{\boldsymbol{\theta}} \left( f_{SC} - \frac{1}{2} \right)^2 \geq \frac{\left\{ Tr\left[ \sigma_{(1,0,\cdots,0)} \cdot \rho_{in} \right] \right\}^2 + \left\{ Tr\left[ \sigma_{(3,0,\cdots,0)} \cdot \rho_{in} \right] \right\}^2}{2^{3+n_c}}, \tag{105}$$

*where we denote*

$$\sigma_{(i_1,i_2,\cdots,i_n)} \equiv \sigma_{i_1} \otimes \sigma_{i_2} \otimes \cdots \otimes \sigma_{i_n},$$

*and the expectation is taken for all parameters in $\boldsymbol{\theta}$ with uniform distributions in $[0, 2\pi]$.*

*Proof.* We expand the function $f_{SC}$ in detail,

$$f_{SC} = \frac{1}{2} + \frac{1}{2}\text{Tr}\left[ \sigma_{(3,0,\cdots,0)} \cdot V_n C X_{n-1} \cdots C X_1 V_1 \cdot \rho_{in} \cdot V_1^\dagger C X_1^\dagger \cdots C X_{n-1}^\dagger V_n^\dagger \right]$$

$$= \frac{1}{2} + \frac{1}{2}\text{Tr}\left[ \sigma_{3,0,\cdots,0} \cdot \rho^{(n)} \right],$$

where

$$\rho^{(j)} = V_j C X_{j-1} \cdots C X_1 V_1 \rho_{in} \cdot V_1^\dagger C X_1^\dagger \cdots C X_{j-1}^\dagger V_j^\dagger, \ \forall j \in [n]. \tag{106}$$

Now we focus on the expectation of $(f_{SC} - \frac{1}{2})^2$:

$$\mathbb{E}_{\boldsymbol{\theta}} \left( f_{SC} - \frac{1}{2} \right)^2 = \mathbb{E}_{\boldsymbol{\theta}_1} \mathbb{E}_{\boldsymbol{\theta}_2} \cdots \mathbb{E}_{\boldsymbol{\theta}_n} \frac{1}{4}\text{Tr}\left[ \sigma_{3,0,\cdots,0} \cdot \rho^{(n)} \right]^2 \tag{107}$$

$$\geq \mathbb{E}_{\boldsymbol{\theta}_1} \mathbb{E}_{\boldsymbol{\theta}_2} \cdots \mathbb{E}_{\boldsymbol{\theta}_{n-1}} \frac{1}{8}\text{Tr}\left[ \sigma_{1,0,\cdots,0} \cdot \rho^{(n-1)} \right]^2 \tag{108}$$

$$\geq \mathbb{E}_{\boldsymbol{\theta}_1} \mathbb{E}_{\boldsymbol{\theta}_2} \cdots \mathbb{E}_{\boldsymbol{\theta}_{n-n_c}} \frac{1}{2^{2+n_c}}\text{Tr}\left[ \sigma_{1,0,\cdots,0} \cdot \rho^{(n-n_c)} \right]^2 \tag{109}$$

$$= \mathbb{E}_{\boldsymbol{\theta}_1} \mathbb{E}_{\boldsymbol{\theta}_2} \cdots \mathbb{E}_{\boldsymbol{\theta}_{n-n_c-1}} \frac{1}{2^{2+n_c}}\text{Tr}\left[ \sigma_{1,0,\cdots,0} \cdot \rho^{(n-n_c-1)} \right]^2 \tag{110}$$

$$= \mathbb{E}_{\boldsymbol{\theta}_1} \frac{1}{2^{2+n_c}}\text{Tr}\left[ \sigma_{1,0,\cdots,0} \cdot \rho^{(1)} \right]^2 \tag{111}$$

$$= \frac{1}{2^{3+n_c}} \left( \text{Tr}\,[\sigma_{1,0,\cdots,0} \cdot \rho_{in}]^2 + \text{Tr}\,[\sigma_{3,0,\cdots,0} \cdot \rho_{in}]^2 \right), \tag{112}$$

where Eq. (112) is derived from Lemma C.3. We derive Eqs. (108-109) by noticing that following equations hold for $n - n_c + 1 \leq j \leq n$ and $i \in \{1,3\}$,

$$\mathbb{E}_{\boldsymbol{\theta}_j} \text{Tr}\left[ \sigma_{i,0,\cdots,0} \cdot \rho^{(j)} \right]^2 = \mathbb{E}_{\boldsymbol{\theta}_j} \text{Tr}\left[ \sigma_{i,0,\cdots,0} \cdot V_j C X_{j-1} \rho^{(j-1)} C X_{j-1}^\dagger V_j^\dagger \right]^2 \tag{113}$$

$$= \frac{1}{2}\text{Tr}\left[ \sigma_{1,0,\cdots,0} \cdot C X_{j-1} \rho^{(j-1)} C X_{j-1}^\dagger \right]^2 \tag{114}$$

$$+ \frac{1}{2}\text{Tr}\left[ \sigma_{3,0,\cdots,0} \cdot C X_{j-1} \rho^{(j-1)} C X_{j-1}^\dagger \right]^2 \tag{115}$$

$$\geq \frac{1}{2}\text{Tr}\left[ \sigma_{1,0,\cdots,0} \cdot C X_{j-1} \rho^{(j-1)} C X_{j-1}^\dagger \right]^2 \tag{116}$$

$$= \frac{1}{2}\text{Tr}\left[ \sigma_{1,0,\cdots,0} \cdot \rho^{(j-1)} \right]^2, \tag{117}$$

where Eq. (113) is derived based on the definition of $\rho^{(j)}$ in Eq. (106), Eqs. (114-115) are derived based on Lemma C.3, and Eq. (117) is derived based on Lemma C.1.

We derive Eq. (110-111) by noticing that following equations hold for $2 \leq j \leq n - n_c$,

$$\mathbb{E}_{\boldsymbol{\theta}_j} \mathrm{Tr} \left[ \sigma_{1,0,\cdots,0} \cdot \rho^{(j)} \right]^2 = \mathbb{E}_{\boldsymbol{\theta}_j} \mathrm{Tr} \left[ \sigma_{1,0,\cdots,0} \cdot V_j C X_{j-1} \rho^{(j-1)} C X_{j-1}^\dagger V_j^\dagger \right]^2 \tag{118}$$

$$= \mathrm{Tr} \left[ \sigma_{1,0,\cdots,0} \cdot C X_{j-1} \rho^{(j-1)} C X_{j-1}^\dagger \right]^2 \tag{119}$$

$$= \mathrm{Tr} \left[ \sigma_{1,0,\cdots,0} \cdot \rho^{(j-1)} \right]^2, \tag{120}$$

where Eq. (118) is based on the definition of $\rho^{(j)}$ in Eq. (106), Eq. (119) is based on Lemma C.3, and Eq. (120) is based on Lemma C.1.

$\square$

