# OpenReview forum: "Toward Trainability of Quantum Neural Networks"
_ICLR.cc/2021/Conference — Reject_

### Official Review · AnonReviewer2 · 2020-10-28
**Review of "Toward Trainability of Quantum Neural Networks"**

**Rating:** 6
**Confidence:** 4

**Review:**

##########################################################################

Summary:


The design of a useful generalization of neural networks on quantum computers has been challenging because the gradient signal will decay exponentially with respect to the depth of the quantum circuit (saturating to exponentially small in system size after the depth is linear in system size). This work provides a detailed analysis of quantum neural networks with a tree structure that uses only a depth logarithmic in the system size. The authors show that the gradient signal will only be polynomially small with respect to the system size. The authors also provide empirical verification of the theoretical analysis showing a much larger gradient norm. However, the improvement in prediction accuracy (under early stopping) when using tree-structure quantum neural networks is not very significant. This is likely because the considered system size (8 qubits) is too small to fully demonstrate the exponential decay and the inability to train random quantum neural networks.

##########################################################################

Reasons for score:

I think the theoretical analysis of the tree-structure quantum neural networks is nice. It provides a rigorous result for training a promising class of quantum neural networks. The proven result can be derived in a straightforward manner if we know that the gradient signal decays exponentially in the depth (which is not what was originally stated in the barren plateau paper [1], the result in [1] only shows an exponential decay in the system size after a depth linear in the system size). This is because a quantum neural network with a tree structure only has a log(n) depth, so the gradient norm would be 2^{-log(n)} = 1/n. However, I don't think this intuition that gradient signal decays exponentially in circuit depth is widely known, so this work still provides a novel contribution from a theoretical aspect.

##########################################################################Pros:

Pros:

1. The barren plateau problem has been a challenge that the quantum machine learning community has to overcome. This work provides a promising class of quantum neural networks that do not suffer from the barren plateau problem.

2. The authors provide rigorous support for the proposed class of quantum neural networks.

3. A good set of numerical experiments supplement their theoretical analysis.


##########################################################################

Cons:


1. The proven result can be derived in a straightforward manner if we know that the gradient signal decays exponentially in the depth.

2.  We do not see a large improvement in prediction accuracy when the random quantum neural network employs early stopping. For example, in fig. 4(a), we can stop at iteration 50~60 and the prediction error of the random quantum neural network would be fairly small.



##########################################################################

Questions and comments during rebuttal period:

1. Lower bound should be $\Omega(\cdots)$ rather than $\mathcal{O}(\cdots)$.


2. The proposed TT-QNN will have a limited expressibility due to the logarithmic depth. Do you know any application where the target function can be represented using a TT-QNN?


3. I would strongly encourage the authors to run experiments with a larger system size (e.g., 12 to 15 qubits). Experiments with 8 qubits are too small and a gradient norm of 0.2 (for random QNN) is quite large. This is likely the cause of the lack of a large prediction advantage when using TT-QNN. Random QNN is just completely untrainable in large system sizes. Numerical experiments with a large system size will likely make the paper much stronger from an empirical perspective (if we can see a larger improvement in prediction error).


[1] McClean, Jarrod R., et al. "Barren plateaus in quantum neural network training landscapes." Nature communications 9.1 (2018): 1-6.

---

> ### Author Response · Authors · 2020-11-25
> **Response to the AnonReviewer2**
>
> We thank the reviewer for the constructive suggestions and feedback, and feel honored for been approbated on the theoretical value of this work! We provide discussions and explanations around the reviewer's concerns as follows.
>
> 1. The proven result can be derived in a straightforward manner if we know that the gradient signal decays exponentially in the depth.
>
> Re: The phenomenon that the gradient signal decays exponentially in the depth is discussed in papers [1,2] and other following works, which employ the (global or local) unitary 2-design assumptions on gates in the circuit.
> Our first concern is that unitary 2-design is in fact a finite gate set that may not suit continuously tuned gates in variational quantum circuits. Thus a consideration could be eliminated by changing the 2-design assumption into tuned gates lie in the unitary space, which in fact do not change the formulation of previous results due to the definition of the 2-design.
> However, tuned gates in the unitary space are hard to implement directly on today's or near-term quantum computers, especially for gates on more than one qubit. Besides, as mentioned at the beginning of Section 3.2, gates in the unitary space could introduce the {RX, RZ} gates and may worsen the performance of QNNs in real-world problems (as shown in Appendix A.4).
> As a conclusion, QNN analysis without the 2-design assumptions is required, which is the main motivation of this work.
>
> On the other hand, we investigate quantum neural networks with other different structures in the newly updated versions, namely the step controlled (SC) QNNs (Figure 2 in Section 3.2). We emphasize that the SC-QNN has a circuit depth linear to the qubit number, however, the corresponding gradient norm is bounded by the term independent from the qubit number (Theorem 3.1). The result of the SC-QNN implies the difference between our theoretical framework and the previous 2-design based frameworks.
>
>
> 2. We do not see a large improvement in prediction accuracy when the random quantum neural network employs early stopping. For example, in fig. 4(a), we can stop at iteration 50~60 and the prediction error of the random quantum neural network would be fairly small. I would strongly encourage the authors to run experiments with larger system size (e.g., 12 to 15 qubits). Experiments with 8 qubits are too small and a gradient norm of 0.2 (for random QNN) is quite large. This is likely the cause of the lack of a large prediction advantage when using TT-QNN. Random QNN is just completely untrainable in large system sizes. Numerical experiments with a large system size will likely make the paper much stronger from an empirical perspective (if we can see a larger improvement in prediction error).
>
>
> Re: Thanks for the constructive suggestions! We implement numerical simulations on different QNNs with qubit number {8,10,12} which are presented in Section 4.2 and Appendix A.2. The results suggest that TT-QNNs and SC-QNNs remain good trainability to qubit size at least up to 12 with test accuracy higher than 0.8. Random QNNs show some sort of trainability for the 8 and 10 qubits case with test accuracy 0.70-0.80 and become totally untrainable when the qubit number is 12.
>
>
>
>
> 3. The proposed TT-QNN will have a limited expressibility due to the logarithmic depth. Do you know any application where the target function can be represented using a TT-QNN?
>
> Re: We notice the TT-QNN structure in the paper [3], in which TT-QNNs are used for 1) real-world problems like classifying MNIST, and 2) classifying quantum states from different physical processes.
>
> 4. Lower bound show be $\Omega()$ rather than $\mathcal{O}()$:
>
> Re: Thanks for pointing out the problem of complexity notations! We have corrected the notation in the newly updated version.
>
>
>
>
> [1]. Grant E, Wossnig L, Ostaszewski M, et al. An initialization strategy for addressing barren plateaus in parametrized quantum circuits. Quantum, 2019, 3: 214.
>
> [2]. Cerezo M, Sone A, Volkoff T, et al. Cost-function-dependent barren plateaus in shallow quantum neural networks. arXiv preprint arXiv:2001.00550, 2020.
>
> [3]. Grant E, Benedetti M, Cao S, et al. Hierarchical quantum classifiers. npj Quantum Information, 2018, 4(1): 1-8.

---

### Official Review · AnonReviewer4 · 2020-10-28
**Toward Trainability of Quantum Neural Networks**

**Rating:** 5
**Confidence:** 4

**Review:**

The authors propose a new analysis of a certain type of trainable quantum circuits, namely Tree Tensor Quantum Neural Networks (TT-QNNs), that gives new and positive guarantees of its learning abilities. Using their methods, that include a specific encoding circuit and a specific loss function, the gradient of each parameter vanishes less badly compared to the usual Random QNNs. The authors demonstrate a lower bound on the (expectation value of the) norm of the overall gradient, which is also observed numerically. This could ensure a better trainability of these quantum circuits, which is of great interest. Their methodology could be used in other circuit designs. The article is well written and quantum computing concepts are well introduced, at least for an accustomed audience.

However, some remarks could cloud the main results.

Major Remarks [concerning the encoding circuit] :
- The encoding circuit seems poorly efficient: it has to be trained for each sample of the dataset. For each sample, the circuit has to be changed n times to scan all measurement configurations. For each configuration, multiple measurements are needed to estimate the gradient of one parameter among the 3xLxn ones (?). And then this has to be repeated through many iterations to reach convergence. Is this procedure realistic? Maybe it is considering that the circuit itself is shallow but repeated many times. A comment from the authors would be useful.
- Worsening the case of the previous question, it seems that each time the encoding circuit is run, it is needed to have the quantum state |x_in>, the amplitude encoding of the sample, as input. How is it done in practice during the training part of each encoding circuit, since the authors themselves say that it seems unplausible to obtain such  states with near term quantum computers?
- p.5 it is said that the simulation has been done "classically" on this part, why? No numerical simulation is shown to see how well the training happened. Is it efficient? Is the input state created close to the amplitude encoding objective?
- Finally, if the encoding circuit proposed in this work was not to be used, and if one would use a random encoding instead, what could be said on the bounds of alpha(ro_in)? Could this kill the benefit claimed and lead to a similar vanishing gradient as Random QNNs?

Other Remarks :
- In the complexity analysis of Algorithm 2, it would have been more informative to provide the runtime O(n_gate n_para n_train T) in terms of n, m, L, s, and other circuit parameters. This makes more sense to grasp the difficulty of the proposed method. And maybe to compare it to the Random QNN Complexity Analysis? At least in the Appendix.
- The choice of the form of the objective function f in (2), and the loss function in (7) should be better explained. Has it been chosen to suit the binary classification set up? and/or to ensure the bound necessary during the proof of Thm.3.1?
- Same question for the choice of Pauli-Z based measurements in both encoding and classifier circuits.
- Similarly, the authors should explain in the main paper the choice of using only RY rotations in the main circuit. It seems from reading the Appendix that it is for avoiding to have a unitary 2-design, is it? Could this restrain the applications, compare to what we see in a lot a (random) QNNs with the mixture of the three types of Rotations?

Minor remarks :
- extra parenthesis in the qubit at the top of p.3
- p.7, is n_test renamed n_teXt and then n_infer? It is hard to follow.

---

> ### Author Response · Authors · 2020-11-25
> **Response to the AnonReviewer4 (1/2)**
>
> We thank the reviewer for the constructive suggestions and feedbacks.
> The main contribution of this work is to investigate the trainability of QNNs. The encoding circuit is a plugin-in part of the main paper, with which we prove additional bounds on the norm of the gradient.
> We would like to provide discussions and explanations around the reviewer's concerns of the encoding circuit as follows.
>
>
> 1. The encoding circuit seems poorly efficient: it has to be trained for each sample of the dataset. For each sample, the circuit has to be changed n times to scan all measurement configurations. For each configuration, multiple measurements are needed to estimate the gradient of one parameter among the 3xLxn ones (?). And then this has to be repeated through many iterations to reach convergence. Is this procedure realistic? Maybe it is considering that the circuit itself is shallow but repeated many times. A comment from the authors would be useful.
>
> Re: We provide a detailed discussion of the encoding circuit and the input model. Specifically, the encoding circuit needs to be trained for each sample. In fact, different samples $x_{in}$ would correspond to different parameters $\beta$ in the encoding circuit.
> However, Algorithm 1 is a $100\\%$ classical algorithm that is written in the circuit language for convenience. Thus, we do not estimate gradients by measurements but calculate them analytically. We have modified some notations in the Section 3.3 (e.g., $W(|{x}_{in}\rangle)$ to $W({\beta})$) for more clear presentations. Theoretical convergence analysis about the encoding circuit can be an interesting direction but is out of this work. We present some experimental details about the encoding circuit training in Appendix A.1.
>
> 2. It seems that each time the encoding circuit is run, it is needed to have the quantum state $|x_{in}>$, the amplitude encoding of the sample, as input. How is it done in practice during the training part of each encoding circuit, since the authors themselves say that it seems unplausible to obtain such states with near term quantum computers?
>
> Re: Since our target in Algorithm 1 is to prepare an approximate amplitude encoding circuit, we do not assume the existence of the amplitude encoding itself. Algorithm 1 takes the classical vector ${x}_{in}$ as the input and output classical parameters ${\beta}$ that could guide a quantum encoding circuit.
>
> 3. Page.5 it is said that the simulation has been done "classically" on this part, why? No numerical simulation is shown to see how well the training happened. Is it efficient? Is the input state created close to the amplitude encoding objective?
>
> Re: A more proper expression is that Algorithm 1 is not developed for running on the quantum computer, but is a classical algorithm that outputs classical parameters that could be used in other quantum algorithms (e.g. the Algorithm 2 in this paper). Thus we do not need to evaluate the quantum efficiency of this procedure.
>
> We present some experimental details about the training in Algorithm 1 in Appendix A.1. As we discussed at the beginning of Section 3.3, convergence analysis on the $l_2$-norm of the amplitude encoding remains a hard open problem.
> Based on the results in Appendix A.1, the training of Algorithm 1 could converge, while the visualization shows that the encoding circuit could only catch a few features from the input data (except image 1 which shows good results). Despite this, we obtain relatively good results on binary classification tasks that employ the mentioned encoding circuit.
>
> 4. Finally, if the encoding circuit proposed in this work was not to be used, and if one would use a random encoding instead, what could be said on the bounds of $\alpha(\rho_{in})$? Could this kill the benefit claimed and lead to a similar vanishing gradient as Random QNNs?
>
> Re: Arbitrary random circuit is not a proper choice for the encoding circuit due to the bad trainability. To discuss the influence of different encoding strategies, we perform some experiments in Appendix A.1 in which the encoding circuit is replaced by the exact amplitude encoding. We provide some results in Figure 8 and Table 2 in Appendix A.1.
> Results show that QNNs using the exact encoding have better accuracy than that of QNNs using the encoding circuit (approximate encoding), which is reasonable since the exact amplitude encoding remains all information of the input data. Although we could not prove an input independent bound on the gradient norm for the exact amplitude encoding case, the gradient norm of the experiments with the exact encoding remains large for proposed QNNs while small for Random-QNNs, which suggests the good trainability of TT-QNNs and SC-QNNs with the exact encoding model.

---

> ### Author Response · Authors · 2020-11-25
> **Response to the AnonReviewer4 (2/2)**
>
> 5. In the complexity analysis of Algorithm 2, it would have been more informative to provide the runtime $O(n_{gate} n_{para} n_{train} T)$ in terms of n, m, L, s, and other circuit parameters. This makes more sense to grasp the difficulty of the proposed method. And maybe to compare it to the Random QNN Complexity Analysis? At least in the Appendix.
>
> Re: The complexity analysis of Algorithm 2 is provided for general QNNs (including random QNNs). We have added some discussions on complexity with a more specific form when the circuit is limited to the TT-QNN or the SC-QNN.
> Specifically, both the TT-QNN and the SC-QNN equipped with the $L$-layer encoding circuit in this work have $\mathcal{O}(nL)$ gates and $\mathcal{O}(n)$ parameters, which lead to the training complexity ${\mathcal{O}}(n_{\text{train}} n^2 LT)$ and the test complexity $\mathcal{O}(n_{\text{test}} n L)$.
>
> 6. The choice of the form of the objective function $f$ in (2), and the loss function in (7) should be better explained. Has it been chosen to suit the binary classification set up? and/or to ensure the bound necessary during the proof of Thm.3.1?
>
> Re: We have added some explanations on page 4 that emphasize the choice of the objective function $f$ is related to easy estimations on the value and the gradient of the function $f$ using quantum circuits. We properly use the loss function $\ell$ that commonly used in the classical machine learning algorithms, which suits the binary classification problems. An explanation is also added (on page 7) to emphasize the efficiency of calculating the value and the gradient of the function $l$ using quantum circuits. Moreover, the objective function $f$ is commonly used in existing quantum machine learning algorithms [1,2]. The formulation of Theorem 3.1 is related to the objective function. However, proving bounds in the Theorem 3.1 does not rely on specific objective functions. Similar bounds could be derived for other objective functions.
>
> 7. Same question for the choice of Pauli-Z based measurements in both encoding and classifier circuits.
>
> Re: PauliZ measurement is commonly used in existing quantum algorithms [1,2]. Specifically, the expectation of the PauliZ measurement is the $p(0)-p(1)$, where $p(0)$ and $p(1)$ denote the probability of outcome $0$ and $1$ when measure the qubit, respectively. Thus, outputs of many quantum algorithms, which are simple qubit measurement results, could be mathematically expressed as the expectation of the PauliZ measurement on the output state.
>
> 8. Similarly, the authors should explain in the main paper the choice of using only RY rotations in the main circuit. It seems from reading the Appendix that it is for avoiding to have a unitary 2-design, is it? Could this restrain the applications, compare to what we see in a lot a (random) QNNs with the mixture of the three types of Rotations?
>
> Re: The main reason for only using RY rotations comes from the consideration that the encoded state for the real-world data lies in the real space. Applying RX or RZ rotations could introduce the imaginary term to the quantum state, which may influence the performance of QNNs for real-world problems.
> Avoiding the unitary 2-design is the by-product of this consideration, which then raises the problem discussed in this paper: analyzing QNNs without the unitary 2-design assumption.
>
> On the other hand, we simulate the TT-QNN and the SC-QNN with single-qubit gates randomly sampling from {RX, RY, RZ} with results presented in Figure 9 and Table 6 in Appendix A.4. The results suggest that these new QNNs perform badly compared to the TT-QNN and the SC-QNN which only employ RY rotations. Thus, our consideration is verified experimentally.
>
>
> 9. Minor remarks: extra parenthesis in the qubit at the top of p.3
> p.7, is $n_{test}$ renamed $n_{teXt}$ and then $n_{infer}$? It is hard to follow.
>
> Re: Thanks for pointing out typos! Minor remarks have been corrected.
>
> [1]. Schuld M, Bocharov A, Svore K M, et al. Circuit-centric quantum classifiers. Physical Review A, 2020, 101(3): 032308.
> [2]. Benedetti M, Lloyd E, Sack S, et al. Parameterized quantum circuits as machine learning models. Quantum Science and Technology, 2019, 4(4): 043001.

---

### Official Review · AnonReviewer1 · 2020-11-01
**This paper has sufficient theoretical analysis but is limited in experimental evaluation.**

**Rating:** 5
**Confidence:** 3

**Review:**

Traditional Quantum Neural Networks suffer from poor trainability and one biggest reason is that the gradient vanishes exponentially with the input qubit number.

The paper proposes Tensor Tree(TT) Circuits based Quantum Neural Network to avoid such problems, a sufficient theoretical analysis was provided.

It proved a training guaranteed lower bound of $\mathcal{O}(1/n)$  on the expectation of the gradient norm on TT and further proved a lower bound for  the expectation of the gradient norm that is independent from the input state.

A binary MNIST experiment was also conducted.

Advantages:
A.	Clarity: The paper is well written and easy to be extended.

B.	Originality: The theoretical analysis is sufficient. This paper proposed a framework which can be employed for analyzing QNNs with other different structures in the future.

Disadvantage:

A. Experiment: Although theoretical analysis is sufficient and convincing, the experiment does not convince me very much:
1. In figure 4d, the blue lines showed that the training error is 10% larger than testing error. This should be analyzed. Does it mean that the chosen test data is not very sufficient to represent the performance of the model?
2. The classification accuracy is not a very good criterion to evaluate the binary classification performance, F1 score would be much appropriate.
3. Only 0-1 classification experiment was conducted, the results of some other pairs should also be provided.
4. I suggest that other tensor network Quantum circuits could also be compared.

B. Limitation: I believe that TT-QNN has better trainability and can solve the gradient vanishing.  This paper only discussed an example structure and it is doubtable to generalize to other structures.

C. This paper (with 20 pages) exceeds the maximum size of ICLR which is 8 pages.

---

> ### Author Response · Authors · 2020-11-25
> **Response to the AnonReviewer1**
>
> We thank the reviewer for the constructive suggestions and feedback and feel honored for been approbated on the theoretical value of this work. We provide discussions and explanations around the reviewer's concerns as follows.
>
> 1. In figure 4d, the blue lines showed that the training error is 10\% larger than testing error. This should be analyzed. Does it mean that the chosen test data is not very sufficient to represent the performance of the model?
>
> Re: The figure 4d in the original paper shows the training error is 10\% larger than the test error on the binary classification task between the image class 0 and 1. We consider the following settings as the potential resources for this phenomenon.
> Firstly, the small-size (100$\times$2) test set used in the original version could be a potential reason, due to the lack of sufficiency for representing the performance of the model. Another reason could be the small batch size along with the small training iteration, which may cause overfitting on a certain subset of the training data. To address these considerations, we make the following improvements in the revised version: 1) we change the batch size from 2 to 20, and 2) we extend test sets from 100 to 400 samples for each class. As shown in Table 4 and Table 5 in Appendix A.3, such a phenomenon has disappeared (for most experiments) or alleviated (for special pairs like (1,4) and (2,4)).
>
>
> 2. The classification accuracy is not a very good criterion to evaluate the binary classification performance, F1 score would be much appropriate.
>
> Re: Thanks for the suggestion. Experiments in the original paper are conducted on the balanced dataset (i.e., different image classes have the same size of training or test sets), so we simply analyze the error (accuracy) of different QNNs. To better evaluate the performances of QNNs, we count the F1-score for classification tasks in the newly updated version as well. Table 1 in Section 4.2 and Table 4,5 in Appendix A.3 provide an example of results, in which the F1-score on the test set is calculated. The TT-QNN and the SC-QNN show higher accuracy and F1-score than the Random-QNN for all class pairs.
>
>
>
>
> 3. Only 0-1 classification experiment was conducted, the results of some other pairs should also be provided.
>
> Re: Thanks for the suggestion. We have added results of experiments with different class pairs $(i,j) \in \{0,1,2,3,4\}$ at Table 1, Section 4.2 and Table 4,5 in Appendix A.3. Results suggest that the performance of TT-QNNs and SC-QNNs are better than that of Random-QNNs on different pairs of image class.
>
>
>
> 4. I suggest that other tensor network Quantum circuits could also be compared.
>
> Re: Thanks for the suggestion. We have added the analysis on the step controlled quantum neural network (SC-QNN) (Figure 2, Section 3.2) in the revised version. The numerical simulation on binary classification tasks for TT-QNNs, SC-QNNs, and Random-QNNs are presented in Figure 5, Table 1 in Section 4.2 and Table 4,5 in Appendix A.3. Results show that the performance of TT-QNNs and SC-QNNs is better than that of Random-QNNs in terms of test accuracy, training accuracy, and F1-score.
>
>
>
>
> 5.  I believe that TT-QNN has better trainability and can solve the gradient vanishing. This paper only discussed an example structure and it is doubtable to generalize to other structures.
>
> Re: As mentioned in previous responses, we present a new class of quantum neural networks, namely, the SC-QNN, with theoretical analysis (in Section 3.2 and Appendix F) and numerical experiments (in Section 4.2 and Appendix A). Both theoretical and experimental results suggest the good trainability of SC-QNNs.
>
> 6. This paper (with 20 pages) exceeds the maximum size of ICLR which is 8 pages.
>
> Re: Our main text of the original paper does not exceed the ICLR paper limitation (8 pages). The limitation on the ICLR submission is the 8 pages size for the main paper which does not conclude the reference and the appendix parts. There is no limitation on the page size of the reference or the appendix. Besides, the limitation on the size of the main text of the rebuttal paper is 9 pages, and the main text of our revised version does not exceed this limitation as well.

---

### Decision · Program_Chairs · 2021-01-07
**Final Decision**

**Decision:**

Reject

**Comment:**

This paper introduces two new quantum neural networks with specific structures: TT-QNNs and SC-QNNs. The main contribution of this work is to show a theoretical lower bound that the gradient of the two neural networks (at random initialization) with respect to certain training objectives is well lower bounded by 2^{-2 L}, where L is the number of layers in the network. Previously, the known work only manage to prove this lower bound with less-realistic QNNs with 2-design, or prove an 2^{-poly(n)} lower bounds for random QNNs, where the input of the neural network is an n-qubit. This paper makes a first step towards solving the vanishing gradient problem of QNNs at random initialization.



The major concern of the paper is the usefulness of these QNNs with proposed architectures: The proposed QNNs might be theoretically easier to train, but what if they can only learn a significantly smaller class of functions? In classical world, such phenomenons are very common: Linear classifiers (or even linear functions over prescribed feature mappings) are much easier to train and have much better theoretical properties, but they fail short in terms of representation power comparing to real neural networks.



In this paper, on the theory side, there is no argument about the representation power of these QNNs: It is unclear which set of functions they can represent efficiently, which limits their theoretical interests to machine learning committee. On the empirical side, the reviewers all agree that the empirical results are weak at this point: The proposed new QNNs did not show significant advantages over random QNNs (especially with early stopping), and other types of QNNs were not compared. Moreover, there seems to be some efficiency issue regarding implementing these QNNs -- More convincing empirical evidence or theoretical evidence about the power of these QNNs need to be addressed.